# Mode Connectivity in Auction Design

**Christoph Hertrich**[*]
Department of Mathematics
London School of Economics and Political Science, UK
`c.hertrich@lse.ac.uk`

**Yixin Tao**
ITCS, Key Laboratory of Interdisciplinary Research of Computation and Economics
Shanghai University of Finance and Economics, China
`taoyixin@mail.shufe.edu.cn`

**László A. Végh**
Department of Mathematics
London School of Economics and Political Science, UK
`l.vegh@lse.ac.uk`

## Abstract

Optimal auction design is a fundamental problem in algorithmic game theory. This problem is notoriously difficult already in very simple settings. Recent work in differentiable economics showed that neural networks can efficiently learn known optimal auction mechanisms and discover interesting new ones. In an attempt to theoretically justify their empirical success, we focus on one of the first such networks, *RochetNet*, and a generalized version for *affine maximizer auctions*. We prove that they satisfy *mode connectivity*, i.e., locally optimal solutions are connected by a simple, piecewise linear path such that every solution on the path is almost as good as one of the two local optima. Mode connectivity has been recently investigated as an intriguing empirical and theoretically justifiable property of neural networks used for prediction problems. Our results give the first such analysis in the context of differentiable economics, where neural networks are used directly for solving non-convex optimization problems.

## 1 Introduction

Auction design is a core problem in mechanism design, with immense applications in electronic commerce (such as sponsored search auctions) as well as in the public sector (such as spectrum auctions). In a revenue maximizing auction, the auctioneer needs to design a mechanism to allocate resources to buyers, and set prices in order to maximize the expected revenue. The buyers' preferences are private and they may behave strategically by misreporting them. For this reason, it is often desirable to devise *dominant strategy incentive compatible (DSIC)* and *individually rational (IR)* mechanisms. By definition, in a DSIC mechanism, it is a dominant strategy for the buyers to report the true valuations; in an IR mechanism, each participating truthful buyer receives a nonnegative payoff.

We focus on DSIC and IR mechanisms that maximize the expected revenue, assuming that the buyers' preferences are drawn from a distribution known to the auctioneer. A classical result of

---

[*]Moved to Université Libre de Bruxelles, Belgium, and Goethe-Universität Frankfurt, Germany, after submission of this article.

37th Conference on Neural Information Processing Systems (NeurIPS 2023).

Myerson [1981] provides the optimal mechanism for the case of a single item and arbitrary number of buyers. Finding the optimal mechanisms for more general settings is a tantalizingly difficult problem. We refer the reader to the surveys by Rochet and Stole [2003], Manelli and Vincent [2007] and Daskalakis [2015] for partial results and references. In particular, no analytic solution is known even for two items and two buyers. Selling multiple items to a single buyer is computationally intractable [Daskalakis et al., 2014]. Already for two items and a single buyer, the description of the optimal mechanism may be uncountable [Daskalakis et al., 2013]. Recent work gives a number of important partial characterizations, e.g. Daskalakis et al. [2015], Giannakopoulos and Koutsoupias [2014], as well as results for weaker notions of Bayesian incentive compatibility, e.g. Cai et al. [2012b,a, 2013], Bhalgat et al. [2013].

Conitzer and Sandholm [2002, 2004] proposed the approach of *automated mechanism design* to use optimization and computational methods to obtain (near) optimal mechanisms for specific problems; see also Sandholm and Likhodedov [2015]. An active recent area of research uses machine learning tools. In particular, Dütting et al. [2019] designed and trained neural networks to automatically find optimal auctions. They studied two network architectures, and showed that several theoretically optimal mechanisms can be recovered using this approach, as well as interesting new mechanisms can be obtained. The first network they studied is *RochetNet*. This is a simple two-layer neural network applicable to the single buyer case, leveraging Rochet's [1987] characterization of the optimal mechanism. The second network, *RegretNet* does not require such a characterization and is applicable for multiple buyers; however, it only provides approximate incentive compatibility.

Dütting et al. [2019] coined the term *'differentiable economics'* for this approach, and there has been significant further work in this direction. These include designing auctions for budget constrained buyers [Feng et al., 2018]; multi-facility location Golowich et al. [2018]; balancing fairness and revenue objectives [Kuo et al., 2020]; incorporating non-linear utility functions and other networks trained from interaction data [Shen et al., 2019][2]; designing revenue-maximizing auctions with differentiable matchings [Curry et al., 2022a]; contextual auction design [Duan et al., 2022]; designing taxation policies [Zheng et al., 2022], and more.

The purpose of this work is to supply theoretical evidence behind the success of neural networks in differentiable economics. The revenue is a highly non-convex function of the parameters in the neural network. Curiously, gradient approaches seem to recover globally optimal auctions despite this non-convexity. Similar phenomena have been studied more generally in the context of deep networks, and theoretical explanations have been proposed, in particular, overparametrization [Allen-Zhu et al., 2019, Du et al., 2019].

**Mode connectivity** Recent work has focused on a striking property of the landscape of loss functions of deep neural networks: local optimal solutions (modes) found by gradient approaches are connected by simple paths in the parameter space. We provide an informal definition of *mode connectivity* here.

**Definition 1** ($\varepsilon$-mode connected (informal))**.** *We say that two solutions are $\varepsilon$-mode connected, if they are connected by a continuous path of solutions, such that the loss function does not worsen by more than $\varepsilon$ compared to one of the end points on the entire path.*

This phenomenon was identified by Garipov et al. [2018] and by Draxler et al. [2018]. Mode connectivity can help to explain the empirical performance of stochastic gradient descent (sgd) (or ascent, in case of revenue maximization). To some extent, mode connectivity prevents a poor local minimal valley region on the function value, from which the sgd method cannot escape easily. Suppose such a bad local minimum exists. Then mode connectivity implies that there exists a path from this bad local minimum to a global minimum on which the loss function does not significantly increase. Therefore, the intuition is that from every bad local minimum, a (stochastic) gradient method would eventually be able to find a way to escape. However, we would like to emphasize that mode connectivity does not provide a formal proof of the success of sgd. It only provides a useful intuition of why sgd does not completely trapped in local optima.

Kuditipudi et al. [2019] gave strong theoretical arguments for mode connectivity. They introduce the notion of $\varepsilon$-*dropout stability*: solutions to a neural network such that in each layer, one can remove

---

[2]MenuNet, developed by [Shen et al., 2019], also encodes menu items as RochetNet. However, unlike RochetNet's approach of repeatedly sampling valuations from the underlying distribution, MenuNet discretizes the buyer's valuation space into discrete values.

at least half the neurons and rescale the remaining units such that the loss function increases by at most $\varepsilon$. Solutions that are $\varepsilon$-dropout stable are then shown to be $\varepsilon$-mode connected. Moreover, they show that *noise stability* (see e.g., Arora et al. [2018]) implies dropout stability, and hence, mode connectivity. Nguyen [2019] showed mode connectivity when there is a hidden layer larger than the training dataset. Shevchenko and Mondelli [2020] shows that stochastic gradient descent solutions to sufficiently overparametrized neural networks are dropout stable even if we only keep a small, randomly sampled set of neurons from each layer.

## 1.1 Our contributions

**RochetNet**    In this paper, we first establish mode connectivity properties of *RochetNet*, the architecture in Dütting et al. [2019] for multiple items and a single buyer. These networks have a single hidden layer, corresponding to the menu. Within this hidden layer, each neuron directly corresponds to an *option* in the menu: which contains an allocation and a price offered to the buyer (see Figure 1). The buyer is assigned the single option, including the allocation and the price, maximizing the buyer's utility. Such an option is called *active* for the buyer. The loss function of *RochetNet* is the revenue, the expected price paid by the buyer. Despite its simplicity, the experiments on *RochetNet* in Dütting et al. [2019] gave impressive empirical results in different scenarios. For example, in the experiments with up to six items and uniform value distributions, *RochetNet* achieves almost the same revenue ($99.9\%$) as the Straight-Jacket Auctions in Giannakopoulos and Koutsoupias [2018], which are known to be optimal in this case. This success is not limited to a single example, as *RochetNet* also consistently performs well in other scenarios, including when infinite menu size is necessary. Furthermore, Dütting et al. [2019] demonstrated the usefulness of *RochetNet* in discovering optimal auctions in situations that were previously unexplored from a theoretical perspective.

First, in Theorem 9, we show that for linear utilities, $\varepsilon$-mode connectivity holds between two solutions that are $\varepsilon$-*reducible*: out of the $K+1$ menu options (neurons), there exists a subset of at most $\sqrt{K+1}$ containing an active option for the buyer with probability at least $1 - \varepsilon$. Assuming that the valuations are normalized such that the maximum valuation of any buyer is at most one, it follows that if we remove all other options from the menu, at most $\varepsilon$ of the expected revenue is lost. The assumption of being $\varepsilon$-reducible is stronger than $\varepsilon$-dropout stability that only drops a constant fraction of the neurons. At the same time, experimental results in Dütting et al. [2019] show evidence of this property being satisfied in practice. They experimented with different sized neural networks in a setting when the optimal auction requires infinite menu size. Even with $10,000$ neurons available, only $59$ options were active, i.e., used at least once when tested over a large sample size. We note that this property also highlights an advantage of *RochetNet* over *RegretNet* and other similar architectures: instead of a black-box neural network, it returns a compact, easy to understand representation of a mechanism.

Our second main result (Theorem 10) shows that for $n$ items and linear utilities, if the number of menu options $K$ is sufficiently large, namely, $(2/\varepsilon)^{4n}$, then $\varepsilon$-mode connectivity holds between *any* two solutions for *any* underlying distribution. The connectivity property holds pointwise: for any particular valuation profile, the revenue may decrease by at most $\varepsilon$ along the path. A key tool in this $\varepsilon$-mode connectivity result is a discretization technique from Dughmi et al. [2014]. We note that such a mode connectivity result can be expected to need a large menu size. In Appendix C, we present an example with two disconnected local maxima for $K = 1$.

**Affine Maximizer Auctions**    We also extend our results and techniques to neural networks for affine maximizer auctions (AMA) studied in Curry et al. [2022b]. This is a generalization of *RochetNet* for multi-buyer scenarios. It can also be seen as a weighted variant of the Vickrey–Clarke–Groves (VCG) mechanism [Vickrey, 1961, Clarke, 1971, Groves, 1973]. AMA offers various allocation options. For a given valuation profile, the auctioneer chooses the allocation with the highest weighted sum of valuations, and computes individual prices for the buyers; the details are described in Section 2.2. AMA is DSIC and IR, however, is not rich enough to always represent the optimal auction.

For AMA networks, we show similar results (Theorem 13 and 14) as for *RochetNet*. We first prove $\varepsilon$-mode connectivity holds between two solutions that are $\varepsilon$-*reducible* (see Definition 11). Curry et al. [2022b] provides evidence of this property being satisfied in practice, observing (in Sec. 7.1) *"Moreover, we found that starting out with a large number of parameters improves performance, even though by the end of training only a tiny number of these parameters were actually used."*. Secondly, we also show that if the number of menu options $K$ is sufficiently large, namely, $(16m^3/\varepsilon^2)^{2nm}$,

then $\varepsilon$-mode connectivity holds pointwise between *any* two solutions. That is, it is valid for any underlying distribution of valuations, possibly correlated between different buyers.

**Relation to previous results on mode connectivity**  Our results do not seem to be deducible from previous mode connectivity results, as we outline as follows. Previous literature on mode connectivity investigated neural networks used for prediction. The results in Kuditipudi et al. [2019], Nguyen [2019], Shevchenko and Mondelli [2020] and other papers crucially rely on the properties that the networks minimize a convex loss function between the predicted and actual values, and require linear transformations in the final layer. *RochetNet* and AMA networks are fundamentally different. The training data does not come as labeled pairs and these network architectures are built directly for solving an optimization problem. For an input valuation profile, the loss function is the negative revenue of the auctioneer. In *RochetNet*, this is obtained as the negative price of the utility-maximizing bundle; for AMA it requires an even more intricate calculation. The objective is to find parameters of the neural network such that the expected revenue is as large as possible. The menu options define a piecewise linear surface of utilities, and the revenue in *RochetNet* can be interpreted as the expected bias of the piece corresponding to a randomly chosen input.

Hence, the landscape of the loss function is fundamentally different from those analyzed in the above mentioned works. The weight interpolation argument that shows mode-connectivity from dropout stability is not applicable in this context. The main reason is that the loss function is not a simple function of the output of the network, but is defined by choosing the price of the argmax option. We thus need a more careful understanding of the piecewise linear surfaces corresponding to the menus.

**Significance for Practitioners**  We see the main contribution of our paper in *explaining* the empirical success and providing theoretical foundations for already existent practical methods, and not in inventing new methods. Nevertheless, two insights a practitioner could use are as follows: (i) It is worth understanding the structure of the auction in question. If one can, e.g., understand whether $\varepsilon$-reducibility holds for a particular auction, this might indicate whether RochetNet or AMA are good methods to apply to this particular case. (ii) Size helps: If one encounters bad local optima, increasing the menu size and rerunning RochetNet or AMA might be a potential fix and will eventually lead to a network satisfying mode connectivity.

## 2  Auction Settings

We consider the case with $m$ buyers and one seller with $n$ divisible items each in unit supply. Each buyer has an additive valuation function $v_i(S) := \sum_{j \in S} v_{ij}$, where $v_{ij} \in V$ represents the valuation of the buyer $i$ on item $j$ and $V$ is the set of possible valuations. Throughout the paper, we normalize the range to the unit simplex: we assume $V = [0, 1]$, and $\|v_i\|_1 = \sum_j v_{ij} \leq 1$ for every buyer $i$. With slight abuse of notation, we let $v = (v_{11}, v_{12}, \cdots, v_{ij}, \cdots, v_{mn})^\top$ and $v_i = (v_{i1}, v_{i2}, \cdots, v_{in})^\top$. The buyers' valuation profile $v$ is drawn from a distribution $F \in \mathcal{P}(V^{m \times n})$. Throughout, we assume that the buyers have *quasi-linear utilities*: if a buyer with valuation $v_i$ receives an allocation $x \in [0, 1]^n$ at price $p$, their utility is $v_i^\top x - p$.

The seller has access to samples from the distribution $F$, and wants to sell these items to the buyers through a DSIC[3] and IR auction and maximize the expected revenue. In the auction mechanism, the $i$-th bidder reports a *bid* $b_i \in [0, 1]^n$. The entire bid vector $b \in [0, 1]^{m \times n}$ will be denoted as $b = (b_1, \ldots, b_m) = (b_i, b_{-i})$, where $b_{-i}$ represents all the bids other than buyer $i$. In a DSIC mechanism, it is a dominant strategy for the agents to report $b_i = v_i$, i.e., reveal their true preferences.

**Definition 2** (DSIC and IR auction)**.** *An auction mechanism requires the buyers to submit bids $b_i \in [0, 1]^n$, and let $b = (b_1, \ldots, b_m)$. The output is a set of allocations $x(b) = (x_1(b), \ldots, x_m(b))$, $x_i(b) \in [0, 1]^n$, and prices $p(b) = (p_1(b), \ldots, p_m(b)) \in \mathbb{R}^m$. Since there is unit supply of each item, we require $\sum_i x_{ij}(b) \leq 1$, where $x_{ij}(b)$ is the allocation of buyer $i$ of item $j$.*

---

[3]There exists a weaker notion of incentive compatibility: *Bayesian incentive compatible* (BIC). In a BIC mechanism, it is a dominant strategy for the buyers to report the true valuations if all other buyers also report truthfully. This paper focuses on DISC mechanisms, as DSIC is considered to be more robust than BIC, which assumes a common knowledge of buyers' distributions on valuations.

*(i) An auction is* dominant strategy incentive compatible (DSIC) *if* $v_i^\top x_i(v_i, b_{-i}) - p_i(v_i, b_{-i}) \geq v^\top x_i(b_i, b_{-i}) - p_i(b_i, b_{-i})$ *for any buyer $i$ and any bid $b = (b_i, b_{-i})$.*

*(ii) An auction is* individually rational (IR) *if* $v_i^\top x_i(v_i, b_{-i}) - p_i(v_i, b_{-i}) \geq 0$.

The revenue of a DSIC and IR auction is

$$\texttt{Rev} = \mathbb{E}_{v \sim F} \left[ \sum_i p_i(v) \right] .$$

## 2.1 Single Buyer Auctions: RochetNet

Dütting et al. [2019] proposed RochetNet as a DISC and IR auction for the case of a single buyer. We omit the subscript $i$ for buyers in this case. A (possibly infinite sized) *menu $M$* comprises a set of *options* offered to the buyer: $M = \{(x^{(k)}, p^{(k)})\}_{k \in \mathcal{K}}$. In each option $(x^{(k)}, p^{(k)})$, $x^{(k)} \in [0,1]^n$ represents the amount of items, and $p^{(k)} \in \mathbb{R}_+$ represents the price. We assume that $0 \in \mathcal{K}$, and $(x^{(0)}, p^{(0)}) = (\mathbf{0}, 0)$ to guarantee IR. We call this the *default option*, whereas all other options are called *regular options*. We will use $K$ to denote the number of regular options; thus, $|\mathcal{K}| = K + 1$.

A buyer submits a bid $b \in [0,1]^n$ representing their valuation, and is assigned to option $k(b) \in \mathcal{K}$ that maximizes the utility[4]

$$k(b) \in \arg\max_{k \in \mathcal{K}} b^\top x^{(k)} - p^{(k)} .$$

This is called the *active option* for the buyer. Note that option $0$ guarantees that the utility is nonnegative, implying the IR property. It is also easy to see that such an auction is DSIC. Therefore, one can assume that $b = v$, i.e., the buyer submits their true valuation; or equivalently, the buyer is allowed to directly choose among the menu options one that maximizes their utility. Moreover, it follows from Rochet [1987] that every DSIC and IR auction for a single buyer can be implemented with a (possibly infinite size) menu using an appropriate tie-breaking rule.

Given a menu $M$, the revenue is defined as

$$\texttt{Rev}(M) = \mathbb{E}_{v \sim F} \left[ p^{(k(v))} \right] .$$

**RochetNet** RochetNet (see Figure 1) is a neural network with three layers: an input layer ($n$ neurons), a middle layer ($K$ neurons), and an output layer (1 neuron):

1. the input layer takes an $n$-dimensional bid $b \in V^n$, and sends this information to the middle layer;

2. the middle layer has $K$ neurons. Each neuron represents a regular option in the menu $M$, which has parameters $x^{(k)} \in [0,1]^n$ and $p^{(k)} \in \mathbb{R}_+$, where $x^{(k)} \in [0,1]^n$ represents the allocation of option $k$ and $p^{(k)}$ represents the price of option $k$. Neuron $k$ maps from $b \in V^n$ to $b^\top x^{(k)} - p^{(k)}$, i.e., the utility of the buyer when choosing option $k$;

3. the output layer receives all utilities from different options and maximizes over these options and $0$: $\max\{\max_k\{(x^{(k)})^\top b - p^{(k)}\}, 0\}$.

We will use $\texttt{Rev}(M)$ to denote the revenue of the auction with menu options $\mathcal{K} = \{0, 1, 2, \ldots, K\}$, where $0$ represents the default option $(\mathbf{0}, 0)$.

The training objective for the RochetNet is to maximize the revenue $\texttt{Rev}(M)$, which is done by stochastic gradient ascent. Note, however, that the revenue is the price of an *argmax* option, which makes it a non-continuous function of the valuations. For this reason, Dütting et al. [2019] use a *softmax*-approximation of the *argmax* as their loss function instead. However, *argmax* is used for testing. In Appendix B, we bound the difference between the revenues computed with these two different activation functions, assuming that the probability density function of the distribution $F$ admits a finite upper bound. Lemma 19 shows that the difference between the revenues for *softmax* and *argmax* is roughly inverse proportional to the parameter $Y$ of the *softmax* function. This allows the practitioner to interpolate between smoothness of the loss function and provable quality of the softmax approximation by tuning the parameter $Y$.

---

[4]We assume that ties are broken in favor of higher prices, but it is not hard to see that our results transfer to other tie-breaking rules, too. See also the discussion in Section B.2 of Babaioff et al. [2022].

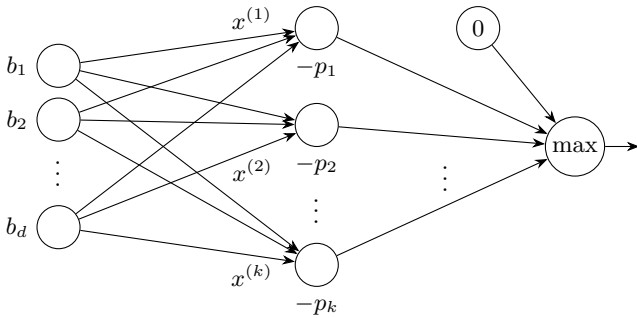

Figure 1: RochetNet: this architecture maps the bid $b$ to the utility of the buyer.

## 2.2 Affine Maximizer Auctions

*Affine Maximizer Auctions (AMA)* also provide a menu $M$ with a set of options $\mathcal{K}$. Each option is of the form $(x^{(k)}, \beta^{(k)}) \in [0,1]^{n \times m} \times \mathbb{R}$, where $x_{ij}^{(k)} \in [0,1]$ represents the allocation of item $i$ to buyer $j$, with the restriction that $\sum_i x_{ij}^{(k)} \leq 1$ for each item $j$, and $\beta^{(k)}$ represents a *'boost'*. We again assume $0 \in \mathcal{K}$, and $(x^{(0)}, \beta^{(0)}) = (\mathbf{0}, 0)$, and call this the *default* option; all other options are called the *regular options*.

Given the bids $b_i \in [0,1]^n$ of the agents, the auctioneer computes a weighted welfare, using weights $w_i \in \mathbb{R}_+$ for the valuations of each agent, and adds the boost $\beta^{(k)}$. Then, the allocation maximizing the weighted boosted welfare is chosen, i.e., the option with

$$k(b) \in \arg\max_{k \in \mathcal{K}} \sum_i w_i b_i^\top x_i^{(k)} + \beta^{(k)}.$$

This will also be referred to as the *active option*. The prices collected from the buyers are computed according to the Vickrey–Clarke–Groves (VCG) scheme. Namely,

$$p_i(b) = \frac{1}{w_i}\left(\sum_{\ell \neq i} w_\ell b_\ell^\top x_\ell^{(k(b_{-i}))} + \beta^{(k(b_{-i}))}\right) - \frac{1}{w_i}\left(\sum_{\ell \neq i} w_\ell b_\ell^\top x_\ell^{(k(b))} + \beta^{(k(b))}\right). \tag{1}$$

Here, $k(b_{-i})$ represents the option maximizing the weighted boosted welfare when buyer $i$ is omitted, i.e., $k(b_{-i}) \in \arg\max_{k \in \mathcal{K}} \sum_{\ell \neq i} w_\ell b_\ell^\top x_\ell^{(k)} + \beta^{(k)}$. It is known that AMA is DSIC and IR. Hence, we can assume that the submitted bids $b_i$ represent the true valuations $v_i$. We also assume the ties are broken in favor of maximizing the total payment. In case of unit weights, this is equivalent to choosing the smallest $\beta^{(k)}$ values, see (2) in Section 4. Given the menu $M$, the revenue of the AMA is

$$\texttt{Rev}(M) = \mathbb{E}_{v \sim F}\left[\sum_i p_i(v)\right].$$

In this paper, we focus on the case when $w_i = 1$ for all buyers. This is also used in the experiments in Curry et al. [2022b]. For this case, AMA can be implemented by a three layer neural network similar to *RochetNet*, with $m \times n$ input neurons. For the more general case when the weights $w_i$ can also be adjusted, one can include an additional layer that combines the buyers' allocations.

Note that for a single buyer and $w_1 = 1$, AMA corresponds to *RochetNet*, with price $p^{(k)} = -\beta^{(k)}$ for each menu option. Indeed, in the formula defining the price $p_i(b)$, the first term is 0, as well as the sum in the second term.

Similarly to *RochetNet*, the loss function, which is maximized via stochastic gradient ascent, is a *softmax*-approximation of the revenue $\texttt{Rev}(M)$, in order to avoid the discontinuities introduced by the *argmax*. We bound the difference in the revenue in Appendix D.3, concluding that it decreases with large parameter $Y$ as in the RochetNet case.

## 2.3 Mode Connectivity

One can view the revenue as a function of the menus, i.e., the parameters in the mechanism: *(i)* in *RochetNet*, $\{(x^{(k)}, p^{(k)})\}_{k \in \mathcal{K}}$; *(ii)* in AMA, $\{(x^{(k)}, \beta^{(k)})\}_{k \in \mathcal{K}}$. We use $\mathcal{M}$ to denote the set of all possible menus.

**Definition 3.** *(Mode connectivity) Two menus $M_1, M_2 \in \mathcal{M}$ are $\varepsilon$-mode-connected if there is a continuous curve $\pi : [0, 1] \to \mathcal{M}$ such that* (i) *$\pi(0) = M_1$;* (ii) *$\pi(1) = M_2$; and* (iii) *for any $t \in [0, 1]$, $\mathtt{Rev}(\pi(t)) \geq \min\{\mathtt{Rev}(M_1), \mathtt{Rev}(M_2)\} - \varepsilon$.*

# 3 Mode Connectivity for the RochetNet

In this section we present and prove our main results for the RochetNet. For some statements, we only include proof sketches. The detailed proofs can be found in Appendix A in the supplementary material. The following definition plays an analogous role to $\varepsilon$-dropout stability in Kuditipudi et al. [2019].

**Definition 4.** *A menu $M$ with $|\mathcal{K}| = K + 1$ options is called $\varepsilon$-reducible if there is a subset $\mathcal{K}' \subseteq \mathcal{K}$ with $0 \in \mathcal{K}'$, $|\mathcal{K}'| \leq \sqrt{K + 1}$ such that, with probability at least $1 - \varepsilon$ over the distribution of the valuation of the buyer, the active option assigned to the buyer is contained in $\mathcal{K}'$.*

As noted in the Introduction, such a property can be observed in the experimental results in Dütting et al. [2019]. The motivation behind this definition is that if a menu satisfies this property, then all but $\sqrt{K + 1}$ options are more or less redundant. In fact, if a menu is $\varepsilon$-reducible, then dropping all but the at most $\sqrt{K + 1}$ many options in $\mathcal{K}'$ results in a menu $M'$ with $\mathtt{Rev}(M') \geq \mathtt{Rev}(M) - \varepsilon$ because the price of any selected option is bounded by $\|v\|_1 \leq 1$.

As a first step towards showing the mode connectivity results, we show that 0-reducibility implies 0-mode-connectivity. We will then use this to derive our two main results, namely that two $\varepsilon$-reducible menus are always $\varepsilon$-mode-connected and that two large menus are always $\varepsilon$-mode-connected.

**Proposition 5.** *If two menus $M_1$ and $M_2$ for the RochetNet are 0-reducible, then they are 0-mode-connected. Moreover, the curve transforming $M_1$ into $M_2$ is piecewise linear with only three pieces.*

To prove Proposition 5, we introduce two intermediate menus $\widehat{M}_1$ and $\widehat{M}_2$, and show that every menu in the piecewise linear interpolation from $M_1$ via $\widehat{M}_1$ and $\widehat{M}_2$ to $M_2$ yields a revenue of at least $\min\{\mathtt{Rev}(M_1), \mathtt{Rev}(M_2)\}$. Using that menu $M_1$ has only $\sqrt{K + 1}$ non-redundant options, menu $\widehat{M}_1$ will be defined by repeating each of the $\sqrt{K + 1}$ options $\sqrt{K + 1}$ times. Menu $\widehat{M}_2$ will be derived from $M_2$ similarly. A technical lemma makes sure that this copying can be done in such a way that each pair of a non-redundant option of $M_1$ and a non-redundant option of $M_2$ occurs exactly for one index in $\widehat{M}_1$ and $\widehat{M}_2$.

To make this more formal, we first assume without loss of generality that $K + 1$ is a square, such that $\sqrt{K + 1}$ is an integer. It is straightforward to verify that the theorem is true for non-squares $K + 1$, too. Suppose the options in $M_1$ and $M_2$ are indexed with $k \in \mathcal{K} = \{0, 1, \ldots, K\}$. Since $M_1$ is 0-reducible, there is a subset $\mathcal{K}_1 \subseteq \mathcal{K}$ with $0 \in \mathcal{K}_1$, $|\mathcal{K}_1| = \sqrt{K + 1}$ such that an option with index in $\mathcal{K}_1$ is selected with probability 1 over the distribution of the possible valuations. Similarly, such a set $\mathcal{K}_2$ exists for $M_2$. To define the curve that provides mode connectivity, we need the following technical lemma, which is proven in Appendix A.

**Lemma 6.** *There exists a bijection $\varphi \colon \mathcal{K} \to \mathcal{K}_1 \times \mathcal{K}_2$ such that for all $k \in \mathcal{K}_1$ we have that $\varphi(k) \in \{k\} \times \mathcal{K}_2$, and for all $k \in \mathcal{K}_2$ we have that $\varphi(k) \in \mathcal{K}_1 \times \{k\}$.*

With this lemma, we can define $\widehat{M}_1$ and $\widehat{M}_2$. Let $\varphi$ the bijection from Lemma 6 and suppose $M_1 = \{(x^{(k)}, p^{(k)})\}_{k \in \mathcal{K}}$. We then define $\widehat{M}_1 = \{(x^{(\varphi_1(k))}, p^{(\varphi_1(k))})\}_{k \in \mathcal{K}}$, where $\varphi_1(k)$ is the first component of $\varphi(k)$. Similarly, $\widehat{M}_2$ is derived from $M_2$ by using the second component $\varphi_2(k)$ of $\varphi(k)$ instead of $\varphi_1(k)$. It remains to show that all menus on the three straight line segments from $M_1$ via $\widehat{M}_1$ and $\widehat{M}_2$ to $M_2$ yield a revenue of at least $\min\{\mathtt{Rev}(M_1), \mathtt{Rev}(M_2)\}$, which is established by the following two propositions; their proofs can be found in Appendix A.

**Proposition 7.** *Let $M = \lambda M_1 + (1-\lambda)\widehat{M}_1$ be a convex combination of the menus $M_1$ and $\widehat{M}_1$. Then $\text{Rev}(M) \geq \text{Rev}(M_1)$. Similarly, every convex combination of the menus $M_2$ and $\widehat{M}_2$ has revenue at least $\text{Rev}(M_2)$.*

The idea to prove Proposition 7 is that, on the whole line segment from $M_1$ to $\widehat{M}_1$, the only active options are those in $\mathcal{K}'$, implying that the revenue does not decrease.

**Proposition 8.** *Let $M = \lambda \widehat{M}_1 + (1-\lambda)\widehat{M}_2$ be a convex combination of the menus $\widehat{M}_1$ and $\widehat{M}_2$. Then, $\text{Rev}(M) \geq \lambda \text{Rev}(\widehat{M}_1) + (1-\lambda)\text{Rev}(\widehat{M}_2)$.*

The idea to prove Proposition 8 is that, due to the special structure provided by Lemma 6, a linear interpolation between the menus also provides a linear interpolation between the revenues. Note that without the construction of Lemma 6, such a linear relation would be false; such an example is shown in Appendix C.

Proposition 5 directly follows from Proposition 7 and Proposition 8. Based on Proposition 5, we can show our two main theorems for the RochetNet. The first result follows relatively easily from Proposition 5.

**Theorem 9.** *If two menus $M_1$ and $M_2$ for the RochetNet are $\varepsilon$-reducible, then they are $\varepsilon$-mode-connected. Moreover, the curve transforming $M_1$ into $M_2$ is piecewise linear with only five pieces.*

*Proof.* We prove this result by showing that every $\varepsilon$-reducible menu $M$ can be linearly transformed into a $0$-reducible menu $\widetilde{M}$ such that each convex combination of $M$ and $\widetilde{M}$ achieves a revenue of at least $\text{Rev}(M) - \varepsilon$. This transformation converting $M_1$ and $M_2$ to $\widetilde{M}_1$ and $\widetilde{M}_2$, respectively, yields the first and the fifth of the linear pieces transforming $M_1$ to $M_2$. Together with Proposition 5 applied to $\widetilde{M}_1$ and $\widetilde{M}_2$ serving as the second to fourth linear piece; the theorem then follows.

To this end, let $M$ be an $\varepsilon$-reducible menu with options indexed by $k \in \mathcal{K}$. By definition, there is a subset $\mathcal{K}' \subseteq \mathcal{K}$ of at most $\sqrt{K+1}$ many options such that, with probability at least $1 - \varepsilon$, the assigned active option is contained in $\mathcal{K}'$. Let $\widetilde{M}$ consist of the same allocations as $M$, but with modified prices. For an option $k \in \mathcal{K}'$, the price $\tilde{p}^{(k)} = p^{(k)}$ in $\widetilde{M}$ is the same as in $M$. However, for an option $k \in \mathcal{K} \setminus \mathcal{K}'$, we set the price $\tilde{p}^{(k)} > 1$ in $\widetilde{M}$ to be larger than the largest possible valuation of any option $\|v\|_1 \leq 1$. It follows that such an option will never be selected and $\widetilde{M}$ is $0$-reducible.

To complete the proof, let us look at the reward of a convex combination $M' = \lambda M + (1-\lambda)\widetilde{M}$. If for a particular valuation $v$ the selected option in $M$ was in $\mathcal{K}'$, then the same option will be selected in $M'$. This happens with probability at least $1 - \varepsilon$. In any other case, anything can happen, but the revenue cannot worsen by more than the maximum possible valuation, which is $\|v\|_1 \leq 1$. Therefore, $\text{Rev}(M) - \text{Rev}(M') \leq \varepsilon \cdot 1 = \varepsilon$, completing the proof. $\square$

**Theorem 10.** *If two menus $M_1$ and $M_2$ for the RochetNet have size at least $\lceil \frac{4}{\epsilon^2} \rceil^{2n}$, then they are $\varepsilon$-connected. Moreover, the curve transforming $M_1$ into $M_2$ is piecewise linear with only five pieces.*

*Proof Sketch.* The full proof can be found in Appendix A.2. The intuition behind this theorem is that if menus are large, then they should contain many redundant options. Indeed, as in the previous theorem, the strategy is as follows. We show that every menu $M$ of size at least $\lceil \frac{4}{\epsilon^2} \rceil^{2n}$ can be linearly transformed into a $0$-reducible menu $\widetilde{M}$ such that each convex combination of $M$ and $\widetilde{M}$ achieves a revenue of at least $\text{Rev}(M) - \varepsilon$. This transformation converting $M_1$ and $M_2$ to $\widetilde{M}_1$ and $\widetilde{M}_2$, respectively, yields the first and the fifth of the linear pieces transforming $M_1$ to $M_2$. Together with Proposition 5 applied to $\widetilde{M}_1$ and $\widetilde{M}_2$ serving as the second to fourth linear piece, the theorem then follows.

However, this time, the linear transformation of $M$ to $\widetilde{M}$ is much more intricate than in the previous theorem. To do so, it is not sufficient to only adapt the prices. Instead, we also change the allocations of the menu options by rounding them to discretized values. This technique is inspired by Dughmi et al. [2014], but non-trivially adapted to our setting. Since the rounding may also modify the active option for each valuation, we have to carefully adapt the prices in order to make sure that for each valuation, the newly selected option is not significantly worse than the originally selected one. Finally, this property has to be proven not only for $\widetilde{M}$, but for every convex combination of $M$ and $\widetilde{M}$.

After the above rounding, the number of possible allocations for any option is bounded by $\lceil\frac{4}{\epsilon^2}\rceil^n$. Out of several options with the same allocation, the buyer would always choose the cheapest one, implying that the resulting menu $\widetilde{M}$ is 0-reducible. $\qquad\square$

## 4 Mode Connectivity for the Affine Maximizer Auctions

Throughout this section, we focus on AMAs with fixed weights $w_i = 1$ for all buyers $i$. Similarly to *RochetNet*, we have the following definition for AMAs.

**Definition 11.** *A menu $M$ with $K + 1$ options is $\varepsilon$-reducible if and only if there exists a subset $\mathcal{K}' \subseteq \mathcal{K}$, $0 \in \mathcal{K}'$, $|\mathcal{K}'| \leq \sqrt{K+1}$ such that, with probability at least $1 - \frac{\varepsilon}{m}$ over the distribution of the valuation of the buyers, (i) $k(v_{-i}) \in \mathcal{K}'$ for any buyer $i$; and (ii) $k(v) \in \mathcal{K}'$.*

Such phenomena are observed in the experiments in [Curry et al., 2022b, Section 6.3].

Our two main results, namely that two $\varepsilon$-reducible menus are always $\varepsilon$-connected and two large menus are always $\varepsilon$-connected, are based on the following proposition, in which we show that 0-reducibility implies 0-connectivity.

**Proposition 12.** *If two menus $M_1$ and $M_2$ are 0-reducible, then they are 0-connected. Moreover, the curve transforming $M_1$ into $M_2$ is piecewise linear with only three pieces.*

The proof idea is similar to the proof of Proposition 5 in *RochetNet*, but requires additional arguments due to the more intricate price structure (see Appendix D.1 for more details). Based on this proposition, now, we are able to show our two main results. First, we achieve $\varepsilon$-connectivity from $\varepsilon$-reducibility.

**Theorem 13.** *If two AMAs $M_1$ and $M_2$ are $\varepsilon$-reducible, then they are $\varepsilon$-mode-connected. Moreover, the curve transforming $M_1$ to $M_2$ is piecewise linear with only five pieces.*

Before the proof, we recall how the total payment is calculated for a valuation profile $v$. We choose $k(v)$ as the option which maximizes the boosted welfare, $\sum_i v_i^\top x_i^{(k)} + \beta^{(k)}$. According to (1), the total revenue can be written as

$$\sum_i p_i(v) = \sum_i \underbrace{\left(\sum_{\ell \neq i} v_\ell^\top x_\ell^{(k(v_{-i}))} + \beta^{(k(v_{-i}))}\right)}_{\text{boosted welfare of } v_{-i}} - (m-1)\underbrace{\left(\sum_i v_i^\top x_i^{(k(v))} + \beta^{(k(v))}\right)}_{\text{boosted welfare of } v} - \beta^{(k(v))}.$$

(2)

*Proof of Theorem 13.* Similar to the proof of Theorem 9, it is sufficient to show that every $\varepsilon$-reducible menu $M$ can be linearly transformed into a 0-reducible menu $\widetilde{M}$ such that each convex combination of $M$ and $\widetilde{M}$ achieves a revenue of at least $\text{Rev}(M) - \varepsilon$. This can then be used as the first and fifth linear piece of the curve connecting $M_1$ and $M_2$, while the middle three pieces are provided by Proposition 12.

We construct $\widetilde{M}$ by *(i)* keeping all options in $\mathcal{K}'$ unchanged; *(ii)* for the options $k \in \mathcal{K} \setminus \mathcal{K}'$, we decrease $\beta^{(k)}$ to be smaller than $-m$, which implies such an option will never be selected (recall that $0 \in \mathcal{K}'$ is assumed, and the option $(\mathbf{0}, 0)$ is better than any such option). Consequently, $\widetilde{M}$ is 0-reducible.

To complete the proof, let us look at the revenue of $M' = \{(x'^{(k)}, \beta'^{(k)})\}_{k \in \mathcal{K}}$, which is a convex combination of $M$ and $\widetilde{M}$: $M' = \lambda M + (1 - \lambda)\widetilde{M}$ for $0 \leq \lambda < 1$. Let $k'(v) = \arg\max_k \sum_i v_i^\top x'^{(k)}_i + \beta'^{(k)}$. As we decrease $\beta^{(k)}$ for $k \notin \mathcal{K}'$, $k(v) \in \mathcal{K}'$ implies $k'(v) \in \mathcal{K}'$ and, additionally, option $k'(v)$ and option $k(v)$ achieve the same boosted welfare and same $\beta$. Therefore, since $M$ is $\varepsilon$-reducible, with probability at least $1 - \frac{\varepsilon}{m}$, the boosted welfare of $v$ as well as the boosted welfare of $v_{-i}$ for all buyers $i$ is the same for $M$ and for $M'$. According to the formula (2), the total payment for the profile $v$ is the same for $M$ and $M'$. Therefore, the loss on the revenue can only appear with probability at most $\frac{\varepsilon}{m}$, and the maximum loss is at most $m$, which implies an $\varepsilon$ loss in total. $\qquad\square$

Second, we show that mode connectivity also holds for those AMAs with large menu sizes, namely for $K + 1 \geq \lceil \frac{16m^3}{\epsilon^2} \rceil^{2nm}$.

**Theorem 14.** *For any $0 < \epsilon \leq \frac{1}{4}$, if two AMAs $M_1$ and $M_2$ have at least $K + 1 \geq \lceil \frac{16m^3}{\epsilon^2} \rceil^{2nm}$ options, then they are $\varepsilon$-mode-connected. Moreover, the curve transforming $M_1$ to $M_2$ is piecewise linear with only five pieces.*

*Proof Sketch.* The full proof can be found in Appendix D.2. Similar to *RochetNet*, the idea of proving Theorem 14 is to discretize the allocations in the menu, then one can use Proposition 12 to construct the low-loss transformation from $M_1$ to $M_2$ by five linear pieces. To do this, one wants the loss of revenue to be small during the discretization. Consider the formula (2) of the total payment. The first two terms do not change much by a small change of the discretization. However, the last term $\beta^{(k(v))}$ might be significantly affected by discretization, which may cause a notable decrease in the total payment. To avoid this, we perform a proportional discount on $\beta$, incentivizing the auctioneer to choose an allocation with a small $\beta$. By this approach, the original revenue will be approximately maintained. Furthermore, we show a linear path, connecting the original menu and the menu after discretizing, which will suffer a small loss. $\square$

## 5 Conclusion

We have given theoretical evidence of mode-connectivity in neural networks designed to learn auction mechanisms. Our results show that, for a sufficiently wide hidden layer, $\varepsilon$-mode-connectivity holds in the strongest possible sense. Perhaps more practically, we have shown $\varepsilon$-mode-connectivity under $\varepsilon$-reducibility, i.e., the assumption that there is a sufficiently small subset of neurons that preserve most of the revenue. There is evidence for this assumption in previous work in differentiable economics. A systematic experimental study that verifies this assumption under various distributions and network sizes is left for future work.

Our results make a first step in providing theoretical arguments underlying the success of neural networks in mechanism design. Our focus was on some of the most basic architectures. A natural next step is to extend the arguments for AMA networks with variable weights $w_i$. Such a result will need to analyze a four layer network, and thus could make headway into understanding the behaviour of deep networks. Besides *RochetNet*, Dütting et al. [2019] also proposed *RegretNet*, based on minimising a regret objective. This network is also applicable to multiple buyers, but only provides approximate incentive compatibility, and has been extended in subsequent work, e.g., Feng et al. [2018], Golowich et al. [2018], Duan et al. [2022]. The architecture is however quite different from *RochetNet*: it involves two deep neural networks in conjunction, an allocation and a payment network, and uses expected ex post regret as the loss function. We therefore expect a mode-connectivity analysis for *RegretNet* to require a considerable extension of the techniques used by us. We believe that such an analysis would be a significant next step in the theoretical analysis of neural networks in differentiable economics.

### Acknowledgments and Disclosure of Funding

All three authors gratefully acknowledge support by the European Research Council (ERC) under the European Union's Horizon 2020 research and innovation programme (for all three authors via grant agreement ScaleOpt–757481; for Christoph Hertrich additionally via grant agreement ForEFront–615640). Yixin Tao also acknowledges the Grant 2023110522 from SUFE.

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
