# Supplemental:
# Mode Connectivity in Auction Design

## A    Detailed Proofs of the Mode Connectivity for the RochetNet

In this section we provide the detailed proofs omitted in Section 3.

### A.1    Interpolating between 0-reducible menus

We start with the proofs of statements on the way towards proving Proposition 5.

*Proof of Lemma 6.* We prove the claim by providing an explicit construction for $\varphi$ in two different cases.

First, suppose that there is some $k^* \in \mathcal{K}_1 \cap \mathcal{K}_2$. In this case, start by setting $\varphi(k) := (k, k)$ for all $k \in \mathcal{K}_1 \cap \mathcal{K}_2$. Then, for all $k \in \mathcal{K}_1 \setminus \mathcal{K}_2$, set $\varphi(k) = (k, k^*)$, and for all $k \in \mathcal{K}_2 \setminus \mathcal{K}_1$, set $\varphi(k) = (k^*, k)$. So far, we have not assigned any pair twice and the two conditions of the lemma are already satisfied, so we can simply assign the remaining pairs arbitrarily.

Second, suppose that $\mathcal{K}_1$ and $\mathcal{K}_2$ are disjoint. Note that for this being possible, $\sqrt{K+1}$ must be at least 2. Pick some distinct $k_1, k_1' \in \mathcal{K}_1$ and $k_2, k_2' \in \mathcal{K}_2$. Set $\varphi(k_1) := (k_1, k_2)$, $\varphi(k_1') := (k_1', k_2')$, $\varphi(k_2) := (k_1', k_2)$, and $\varphi(k_2') := (k_1, k_2')$. Then, for all $k \in \mathcal{K}_1 \setminus \{k_1, k_1'\}$, set $\varphi(k) := (k, k_2)$ and for all $k \in \mathcal{K}_2 \setminus \{k_2, k_2'\}$, set $\varphi(k) := (k_1, k)$. Again, we have not assigned any pair twice and the two conditions of the lemma are already satisfied, so we can simply assign the remaining pairs arbitrarily. $\square$

*Proof of Proposition 7.* We only prove the first statement on $M_1$; the statement on $M_2$ follows analogously. We show that for each possible valuation $v$ of the buyer, the price paid to the seller for menu $M$ is at least as high as in menu $M_1$. Suppose for valuation $v$ that the buyer chooses the $k$-th option in menu $M_1$. Note that we may assume $k \in \mathcal{K}_1$ due to 0-reducibility of $M_1$. By construction of $\varphi$, it follows that $\varphi_1(k) = k$. Therefore, the $k$-th option in $M$ is exactly equal to the $k$-th option in $M_1$. Making use of the fact that ties are broken in favor of larger prices, it suffices to show that the $k$-th option is utility-maximizing in $M$, too.

To this end, let $k' \in \mathcal{K}$ be an arbitrary index. If $M_1 = \{(x^{(k)}, p^{(k)})\}_{k \in \mathcal{K}}$, then the utility of option $k'$ in $M$ is

$$
\begin{aligned}
&v^\top(\lambda x^{(k')} + (1-\lambda)x^{(\varphi_1(k'))}) - (\lambda p^{(k')} + (1-\lambda)p^{(\varphi_1(k'))}) \\
={}& \lambda(v^\top x^{(k')} - p^{(k')}) + (1-\lambda)(v^\top x^{(\varphi_1(k'))} - p^{(\varphi_1(k'))}) \\
\leq{}& \lambda(v^\top x^{(k)} - p^{(k)}) + (1-\lambda)(v^\top x^{(k)} - p^{(k)}) \\
={}& v^\top x^{(k)} - p^{(k)},
\end{aligned}
$$

where the inequality follows because the $k$-th option is utility-maximizing for menu $M_1$. This shows that it is utility-maximizing for menu $M$, completing the proof. $\square$

*Proof of Proposition 8.* The claim is trivial for $\lambda = 0$ or $\lambda = 1$. Therefore, assume $0 < \lambda < 1$ for the remainder of the proof. Again we show that the claim holds pointwise for each possible valuation and therefore also for the revenue. For valuation $v$, let $k_1$ and $k_2$ be the active option assigned to the buyer in $\widehat{M}_1$ and $\widehat{M}_2$, respectively. Note that by construction of the menus $\widehat{M}_1$ and $\widehat{M}_2$ we may assume without loss of generality that $k_1 \in \mathcal{K}_1$ and $k_2 \in \mathcal{K}_2$. Let $k^* := \varphi^{-1}(k_1, k_2)$.

We show that option $k^*$ is utility-maximizing in $M = \{(x^{(k)}, p^{(k)})\}_{k \in \mathcal{K}}$. To this end, we use the notation $\widehat{M}_1 = \{(\hat{x}^{(k)}, \hat{p}^{(k)})\}_{k \in \mathcal{K}}$ and $\widehat{M}_2 = \{(\hat{y}^{(k)}, \hat{q}^{(k)})\}_{k \in \mathcal{K}}$. Let $k' \in \mathcal{K}$ be an arbitrary index.

The utility of option $k'$ in menu $M$ can be bounded as follows:

$$
\begin{aligned}
v^\top x^{(k')} - p^{(k')} &= \lambda(v^\top \hat{x}^{(k')} - \hat{p}^{(k')}) + (1-\lambda)(v^\top \hat{y}^{(k')} - \hat{q}^{(k')}) \\
&\leq \lambda(v^\top \hat{x}^{(k_1)} - \hat{p}^{(k_1)}) + (1-\lambda)(v^\top \hat{y}^{(k_2)} - \hat{q}^{(k_2)}) \\
&= \lambda(v^\top \hat{x}^{(k^*)} - \hat{p}^{(k^*)}) + (1-\lambda)(v^\top \hat{y}^{(k^*)} - \hat{q}^{(k^*)}) \\
&= v^\top x^{(k^*)} - p^{(k^*)},
\end{aligned}
$$

where the inequality in the second line follows because $k_1$ and $k_2$ are utility-maximizing for $\widehat{M}_1$ and $\widehat{M}_2$, respectively, and the equality in the third line follows because, by construction, in menu $\widehat{M}_1$ option $k^*$ is equivalent to option $k_1 = \varphi_1(k^*) \in \mathcal{K}_1$, and similarly in menu $\widehat{M}_2$ option $k^*$ is equivalent to option $k_2 = \varphi_2(k^*) \in \mathcal{K}_2$. This concludes the proof that $k^*$ is utility-maximizing.

With the same reasoning as above, we obtain $p^{(k^*)} = \lambda\hat{p}^{(k_1)} + (1-\lambda)\hat{q}^{(k_2)}$, from which we conclude that the price achieved by the seller in menu $M$ for valuation $v$ is at least as high as the convex combination of the achieved prices for menus $\widehat{M}_1$ and $\widehat{M}_2$. $\qquad\square$

## A.2 Discretizing large menus

This subsection is devoted to providing a detailed proof of Theorem 10. To do so, we will show how to convert any menu $M$ of size at least $\lceil\frac{4}{\epsilon^2}\rceil^{2n}$ into a 0-reducible menu $\widetilde{M}$ such that each convex combination of $M$ and $\widetilde{M}$ achieves a revenue of at least $\texttt{Rev}(M) - \varepsilon$. Without loss of generality, we assume that $M$ has size exactly $K + 1 = \lceil\frac{4}{\epsilon^2}\rceil^{2n}$.

To construct the menu $\widetilde{M}$ satisfying these requirements, we adapt techniques from Dughmi et al. [2014].[5] In general, the idea is to discretize the allocations in the menu by a finite allocation set $S$ (see Definition 15) whose size is at most $\sqrt{K+1} = \lceil\frac{4}{\epsilon^2}\rceil^{n}$. However, because of the discretization, the buyer may choose an option with a much smaller price, providing a lower revenue compared to the original menu. To deal with this, we also decrease the prices on the menu; the decrease is in proportion to the price. Intuitively, this incentives the buyer to choose the option with an originally high price. We show, after this modification, the menu achieves a revenue of at least $\texttt{Rev}(M) - \varepsilon$.

For ease of notation, we will use $\tilde{\varepsilon} := \frac{\varepsilon^2}{4}$ and, therefore, $2\sqrt{\tilde{\varepsilon}} = \varepsilon$.

**Definition 15.** *Let $S$ be a (finite) set of allocations. We say that $S$ is an $\tilde{\varepsilon}$-cover if, for every possible allocation $x$, there exists an allocation $\tilde{x} \in S$ such that for every possible valuation vector $v$ we have that $v^\top x \geq v^\top \tilde{x} \geq v^\top x - \tilde{\varepsilon}$.*

The following proposition shows that one can construct an $\tilde{\varepsilon}$-cover $S$ with size at most $\lceil\frac{4}{\epsilon^2}\rceil^{n}$.

**Proposition 16.** *If $\|v\|_1 \leq 1$, then*

$$
S = \underbrace{\{\tilde{\varepsilon}s\}_{s=0}^{\lfloor\frac{1}{\tilde{\varepsilon}}\rfloor} \times \{\tilde{\varepsilon}s\}_{s=0}^{\lfloor\frac{1}{\tilde{\varepsilon}}\rfloor} \times \cdots \times \{\tilde{\varepsilon}s\}_{s=0}^{\lfloor\frac{1}{\tilde{\varepsilon}}\rfloor}}_{n \text{ terms}}
$$

*is an $\tilde{\varepsilon}$-cover, and $|S| = \lceil\frac{1}{\tilde{\varepsilon}}\rceil^{n} = \lceil\frac{4}{\epsilon^2}\rceil^{n}$.*

*Proof.* For any allocation $x$, we can round it down to $\tilde{x}$, such that $\tilde{x}_j = \lfloor\frac{x_j}{\tilde{\varepsilon}}\rfloor \cdot \tilde{\varepsilon}$. It is not hard to see that $v^\top x \geq v^\top \tilde{x}$. Additionally, the inequality $v^\top \tilde{x} \geq v^\top x - \tilde{\varepsilon}$ follows as the total loss is at most $v^\top(\tilde{x} - x) \leq \|v\|_1\|\tilde{x} - x\|_\infty \leq \tilde{\varepsilon}$. $\qquad\square$

**Construction of $\widetilde{M}$.** Given $S$, we can construct $\widetilde{M}$ as follows. Each option $(x^{(k)}, p^{(k)})$ in menu $M$ is modified to $(\tilde{x}^{(k)}, \tilde{p}^{(k)})$ in menu $\widetilde{M}$, where $\tilde{x}^{(k)}$ is the corresponding allocation of $x^{(k)}$ in $S$ and the price is set to $\tilde{p}^{(k)} = \left(1 - \sqrt{\tilde{\varepsilon}}\right)p^{(k)}$:

$$
\widetilde{M} = \left\{\left(\tilde{x}^{(k)}, \tilde{p}^{(k)}\right)\right\}_{k \in \mathcal{K}}.
$$

---

[5]In their paper, they construct a menu with a finite number of options to approximate the optimal mechanism. The approximation is based on the multiplicative error, and they assume the buyer's valuation is no less than 1.

The following lemma shows that this construction indeed ensures that the reward decreases by at most $\varepsilon$.

**Lemma 17.** *It holds that* $\mathtt{Rev}(\widetilde{M}) \geq \mathtt{Rev}(M) - 2\sqrt{\tilde{\varepsilon}} = \mathtt{Rev}(M) - \varepsilon$.

*Proof.* The following inequalities demonstrate the buyer who chooses option $k$ in menu $M$ will not choose option $k'$ in menu $\widetilde{M}$ such that $p^{(k')} < p^{(k)} - \sqrt{\tilde{\varepsilon}}$.

$$
\begin{aligned}
v^\top \tilde{x}^{(k)} - \left(1 - \sqrt{\tilde{\varepsilon}}\right) p^{(k)} &\geq v^\top x^{(k)} - p^{(k)} - \tilde{\varepsilon} + \sqrt{\tilde{\varepsilon}} p^{(k)} \\
&\geq v^\top x^{(k')} - p^{(k')} - \tilde{\varepsilon} + \sqrt{\tilde{\varepsilon}} p^{(k)} \\
&\geq v^\top \tilde{x}^{(k')} - \left(1 - \sqrt{\tilde{\varepsilon}}\right) p^{(k')} - \tilde{\varepsilon} + \sqrt{\tilde{\varepsilon}}(p^{(k)} - p^{(k')}) \\
&> v^\top \tilde{x}^{(k')} - \left(1 - \sqrt{\tilde{\varepsilon}}\right) p^{(k')}. \quad\quad\quad\quad (3)
\end{aligned}
$$

The first and third inequalities hold by Definition 15 and the second inequality holds as the buyer will choose option $k$ in menu $M_1$.

Therefore, the total loss on the revenue is upper bounded by $\sqrt{\tilde{\varepsilon}} p^{(k)} + \sqrt{\tilde{\varepsilon}} \leq 2\sqrt{\tilde{\varepsilon}}$, as the price satisfies $p^{(k)} \leq 1$. $\qquad\square$

In addition to this property of $\widetilde{M}$ itself, we also need to show the revenue does not drop more than $\varepsilon$ for any menu on the line segment connecting $M$ to $\widetilde{M}$.

**Lemma 18.** *Let* $M' = \lambda M + (1 - \lambda)\widetilde{M}$ *be a convex combination of the menus* $M$ *and* $\widetilde{M}$. *Then,* $\mathtt{Rev}(M') \geq \mathtt{Rev}(M) - 2\sqrt{\tilde{\varepsilon}} = \mathtt{Rev}(M) - \varepsilon$.

*Proof.* Let $M = \{(x^{(k)}, p^{(k)})\}_{k \in \mathcal{K}}$ and $M' = \{(x'^{(k)}, p'^{(k)})\}_{k \in \mathcal{K}}$. Similar to the proof of Lemma 17, we show that the buyer who chooses option $k$ in menu $M$ will not choose option $k'$ in menu $M'$ such that $p^{(k')} < p^{(k)} - \sqrt{\tilde{\varepsilon}}$. This is true by the following (in)equalities. For any $k' \in \mathcal{K}$, we have that

$$
\begin{aligned}
v^\top x'^{(k)} - p'^{(k)} &= \lambda(v^\top x^{(k)} - p^{(k)}) + (1 - \lambda)(v^\top \tilde{x}^{(k)} - \tilde{p}^{(k)}) \\
&> \lambda(v^\top x^{(k')} - p^{(k')}) + (1 - \lambda)(v^\top \tilde{x}^{(k')} - \tilde{p}^{(k')}).
\end{aligned}
$$

The inequality follows by combining (i) $v^\top x^{(k)} - p^{(k)} \geq v^\top x^{(k')} - p^{(k')}$, which is true as the buyer will choose option $k$ in menu $M$; and (ii) $v^\top \tilde{x}^{(k)} - \tilde{p}^{(k)} > v^\top \tilde{x}^{(k')} - \tilde{p}^{(k')}$ from (3).

Similar to the proof of Lemma 17, it follows than that the total loss on the revenue is upper bounded by $2\sqrt{\tilde{\varepsilon}}$. $\qquad\square$

With these lemmas at hand, we can finally prove Theorem 10.

*Proof of Theorem 10.* Applying the transformation described in this section to convert $M_1$ and $M_2$ results in two menus $\widetilde{M_1}$ and $\widetilde{M_2}$, respectively. Since $\widetilde{M_1}$ and $\widetilde{M_2}$ contain at most $\sqrt{K+1} = \lceil \frac{4}{\epsilon^2} \rceil^n$ different allocations and a buyer would always choose the cheapest out of several options with the same allocation, they are $0$-reducible. Applying Proposition 5 to them implies that they are $0$-mode-connected with three linear pieces. Combining these observations with Lemmas 17 and 18 implies that $M_1$ and $M_2$ are $\varepsilon$-connected with five linear pieces. $\qquad\square$

## B  Bounds on the Error of the Softmax Approximation for the Argmax

In the RochetNet, to ensure that the objective is a smooth function, a softmax operation is used instead of the argmax during the training process:

$$
\mathtt{Rev^{softmax}}(M) = \int \sum_{k=1}^{K} p_i \frac{e^{Y(x^{(k)\top}v - p^{(k)})}}{\sum_{k'=1}^{K} e^{Y(x^{(k')\top}v - p^{(k')})}} \mathrm{d}F(v).
$$

Here, $Y$ is a sufficiently large constant. In this section, we will look at the difference between the actual revenue and this softmax revenue.

We would like to assume the density of the valuation distribution is upper bounded by $\mathcal{X} = \max_{v \in [0,1]^n \text{ and } \|v\|_1 \leq 1} f(v)$, which is a finite value. Given this assumption, the following lemma shows that, for any menu $M$ of size $K$, the difference between the actual revenue and the softmax revenue is bounded.

**Lemma 19.** *For any $M$ and $Y \geq 1$,*

$$\left| Rev^{softmax}(M) - Rev(M) \right| \leq \frac{K+1}{Y} \left( (n\mathcal{X} + 1 + \frac{\mathcal{X}}{Y}) \log \frac{Y}{\mathcal{X}} + \mathcal{X} \right).$$

*Proof.* We prove $\texttt{Rev}^{\texttt{softmax}}(M) - \texttt{Rev}(M) \leq \frac{K}{Y}\left( (n\mathcal{X} + 1 + \frac{\mathcal{X}}{Y}) \log \frac{Y}{\mathcal{X}} + \mathcal{X} \right)$. $\texttt{Rev}(M) - \texttt{Rev}^{\texttt{softmax}}(M) \leq \frac{K}{Y}\left( (n\mathcal{X} + 1 + \frac{\mathcal{X}}{Y}) \log \frac{Y}{\mathcal{X}} + \mathcal{X} \right)$ follows by a similar argument.

Let $k(v)$ be the option chosen in menu $M$ when the buyer's valuation is $v$. Then, the difference between these two can be bounded as follows.

$$\texttt{Rev}^{\texttt{softmax}}(M) - \texttt{Rev}(M) \leq \int \sum_{k=0}^{K} (p^{(k)} - p^{(k(v))})^+ \cdot \frac{e^{Y(v^\top x^{(k)} - p^{(k)})}}{\sum_{k'=1}^{K} e^{Y(v^\top x^{(k')} - p^{(k')})}} \mathrm{d}F(v)$$

$$\leq \int \sum_{k=0}^{K} (p^{(k)} - p^{(k(v))})^+ e^{Y(v^\top x^{(k)} - p^{(k)} - v^\top x^{(k(v))} + p^{(k(v))})} \mathrm{d}F(v).$$

Here, $(\cdot)^+ \triangleq \max\{\cdot, 0\}$. Now, we focus on one option $k$, and we will give an upper bound on

$$\int (p^{(k)} - p^{(k(v))})^+ \mathbf{1}_{v^\top x^{(k)} - p^{(k)} + \sigma \geq v^\top x^{(k(v))} - p^{(k(v))} \geq v^\top x^{(k)} - p^{(k)}} \mathrm{d}F(v) \tag{4}$$

for the non-negative parameter $\sigma$, which will be specified later. Note that, it is always true that $v^\top x^{(k(v))} - p^{(k(v))} \geq v^\top x^{(k)} - p^{(k)}$. If $v^\top x^{(k)} - p^{(k)} + \sigma \geq v^\top x^{(k(v))} - p^{(k(v))}$ is not satisfied then $e^{Y(v^\top x^{(k)} - p^{(k)} - v^\top x^{(k(v))} + p^{(k(v))})} \leq e^{-Y\sigma}$. Therefore, if (4) is upper bounded by $\mathcal{C}(\sigma)$, then $\texttt{Rev}_M^{\texttt{softmax}} - \texttt{Rev}(M) \leq (K+1)(\mathcal{C}(\sigma) + (1+\sigma)e^{-Y\sigma})$. [6]

Note that

$$\int (p^{(k)} - p^{(k(v))})^+ \mathbf{1}_{v^\top x^{(k)} - p^{(k)} + \sigma \geq v^\top x^{(k(v))} - p^{(k(v))} \geq v^\top x^{(k)} - p^{(k)}} \mathrm{d}F(v)$$

$$\leq \sigma + \int \sum_{j=1}^{n} v_j (x_j^{(k)} - x_j^{(k(v))})^+ \mathbf{1}_{v^\top x^{(k)} - p^{(k)} + \sigma \geq v^\top x^{(k(v))} - p^{(k(v))} \geq v^\top x^{(k)} - p^{(k)}} \mathrm{d}F(v).$$

The inequality follows as we consider the region of $v$ such that $v^\top x^{(k)} - p^{(k)} + \sigma \geq v^\top x^{(k(v))} - p^{(k(v))}$. Additionally, since $v_j \in [0,1]$,

$$\int v_j (x_j^{(k)} - x_j^{(k(v))})^+ \mathbf{1}_{v^\top x^{(k)} - p^{(k)} + \sigma \geq v^\top x^{(k(v))} - p^{(k(v))} \geq v^\top x^{(k)} - p^{(k)}} \mathrm{d}F(v)$$

$$\leq \int (x_j^{(k)} - x_j^{(k(v))})^+ \mathbf{1}_{v^\top x^{(k)} - p^{(k)} + \sigma \geq v^\top x^{(k(v))} - p^{(k(v))} \geq v^\top x^{(k)} - p^{(k)}} \mathrm{d}F(v).$$

Now we fix all coordinates of valuation $v$ other than coordinate $j$. Note that, the function $v^\top x^{(k(v))} - p^{(k(v))} - v^\top x^{(k)} - p^{(k)}$ is a convex function on $v_j$ and $x_j^{(k)} - x_j^{(k(v))}$ is the negative gradient of this

---

[6] Note that if $p^{(k)} \geq 1 + \sigma$ then $v^\top x^{(k)} - p^{(k)} + (p^{(k)} - 1) \leq v^\top x^{(k(v))} - p^{(k(v))}$ as LHS $\leq 0$ and RHS $\geq 0$. Therefore, if $v^\top x^{(k)} - p^{(k)} + \sigma \geq v^\top x^{(k(v))} - p^{(k(v))}$ is not satisfied, then $(p^{(k)} - p^{(k(v))})^+ e^{Y(v^\top x^{(k)} - p^{(k)} - v^\top x^{(k(v))} + p^{(k(v))})} \leq \max_{\sigma' \geq \sigma}\{(1+\sigma')e^{-Y\sigma'}\}$. Note that $\max_{\sigma' \geq \sigma}\{(1 + \sigma')e^{-Y\sigma'}\} \leq (1+\sigma)e^{-Y\sigma}$ when $Y \geq 1$.

convex function. Since we are looking at the region such that the function $v^\top x^{(k(v))} - p^{(k(v))} - v^\top x^{(k)} - p^{(k)}$ is bounded in $[0, \sigma]$, this direct imply

$$\int_{v_j \in [0,1]} (x_j^{(k)} - x_j^{(k(v))})^+ \mathbf{1}_{v^\top x^{(k)} - p^{(k)} + \sigma \geq v^\top x^{(k(v))} - p^{(k(v))} \geq v^\top x^{(k)} - p^{(k)}} \, dF(v) \leq \mathcal{X}\sigma.$$

This implies $\texttt{Rev}_M^{\texttt{softmax}} - \texttt{Rev}(M) \leq (K+1)(\sigma + n\mathcal{X}\sigma + (1+\sigma)e^{-Y\sigma})$ which is upper bounded by $\frac{K+1}{Y}\left((n\mathcal{X} + 1 + \frac{\mathcal{X}}{Y})\log\frac{Y}{\mathcal{X}} + \mathcal{X}\right)$ by setting $\sigma = \frac{1}{Y}\log\frac{Y}{\mathcal{X}}$. $\qquad\square$

## C   Example: Disconnected Local Maxima

This section shows that the revenue is not quasiconcave on $M$, and in fact it might have disconnected local maxima. Recall that a function $g$ is quasiconcave if and only if, for any $x$, $y$ and $\lambda \in [0, 1]$,

$$g(\lambda x + (1-\lambda)y) \geq \min\{g(x), g(y)\}$$

Hence, quasiconcavity implies 0-mode-connectivity with a single straight-line segment.

We consider the case that there is only one buyer, one item, and one regular option on the menu. Consider the following value distribution $f$:

$$f(x) = \begin{cases} 1.5 & 0 < x \leq \frac{1}{3} + 0.15 \\ 0 & \frac{1}{3} + 0.15 < x \leq \frac{2}{3} + 0.15 \\ 1.5 & \frac{2}{3} + 0.15 < x \leq 1. \end{cases} \tag{5}$$

With this probability distribution, we show the following result. As Figure 2 shows, there are two local maxima so that any continuous curve connecting them has lower revenue than either endpoint. Hence, mode connectivity fails between these two points. We only give a formal proof of the fact that the revenue is not quasiconcave.

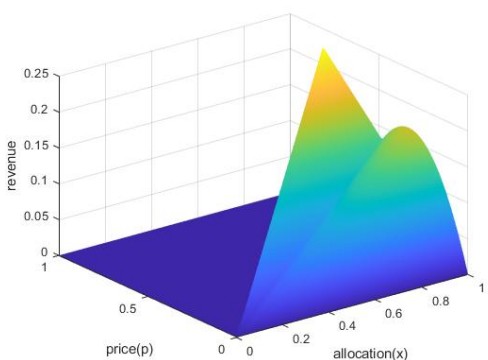

Figure 2: Revenue of the mechanism $M = \{(x, p)\}$ when the value distribution is $f$.

**Lemma 20.** $\texttt{Rev}(M)$ *is not quasiconcave on* $M$.

*Proof.* We consider the case where $n = 1$ (single item case); $K = 1$ (menu with single options). The value distribution, $f$ is defined in (5).

We consider two menus: $M_1$ and $M_2$, where $M_1 = \{(0, 0), (1, 0.36)\}$ and $M_2 = \{(0, 0), (1, 0.84)\}$. Then, $\texttt{Rev}(M_1) = 0.1656$ and $\texttt{Rev}(M_2) = 0.2016$.

However, if we consider $M_3 = \frac{1}{2}(M_1 + M_2) = \{(0, 0), (1, 0.6)\}$, then this provides a revenue of $0.165$, which is strictly smaller than $\texttt{Rev}(M_1)$ and $\texttt{Rev}(M_2)$. More intuitively, Figure 2 shows the revenue for $x \in [0, 1]$ and $p \in [0, 1]$. $\qquad\square$

## D   Detailed Proofs of the Mode Connectivity for AMAs

In this section, we provide the detailed proofs omitted in Section 4

## D.1 Interpolating between 0-reducible menus

In this subsection, we will prove Proposition 12, that is, we show that two 0-reducible menus $M_1 = \{(x^{(1,k)}, \beta^{(1,k)})\}_{k \in \mathcal{K}}$ and $M_2 = \{(x^{(2,k)}, \beta^{(2,k)})\}_{k \in \mathcal{K}}$ are 0-mode-connected.

Similar to *RochetNet*, we introduce two intermediate menus $\widehat{M}_1$ and $\widehat{M}_2$, and show that every menu in the piecewise linear interpolation form $M_1$ via $\widehat{M}_1$ and $\widehat{M}_2$ to $M_2$ yields a revenue of at least $\min\{\texttt{Rev}(M_1), \texttt{Rev}(M_2)\}$. Using that menu $M_1$ has only $\sqrt{K+1}$ non-redundant options, menu $\widehat{M}_1$ will be defined by repeating each of the $\sqrt{K+1}$ options $\sqrt{K+1}$ times. Menu $\widehat{M}_2$ will be derived from $M_2$ similarly.

To make this more formal, let $\mathcal{K}'_1$ ( and $\mathcal{K}'_2$) denote the set of the indexes of options in $M_1$ ( and $M_2$) in definition of $\varepsilon$-reducibility, respectively. Similar to *RochetNet*, with the help of the Lemma 6, we can formally define $\widehat{M}_1$ and $\widehat{M}_2$ as $\widehat{M}_1 = \{(x^{(1,\varphi_1(k))}, \beta^{(1,\varphi_1(k))})\}_{k \in \mathcal{K}}$, where $\varphi_1(k)$ is the first component of $\varphi(k)$; and, similarly, $\widehat{M}_2$ is derived from $M_2$ by using the second component $\varphi_2(k)$ of $\varphi(k)$ instead of $\varphi_1(k)$.

It remains to show that all menus on the three straight line segments from $M_1$ via $\widehat{M}_1$ and $\widehat{M}_2$ to $M_2$ yield revenue of at least $\min\{\texttt{Rev}(M_1), \texttt{Rev}(M_2)\}$.

**Proposition 21.** *Let $M = \lambda M_1 + (1-\lambda)\widehat{M}_1$ be a convex combination of the menus $M_1$ and $\widehat{M}_1$. Then $\texttt{Rev}(M) \geq \texttt{Rev}(M_1)$. Similarly, every convex combination of the menus $M_2$ and $\widehat{M}_2$ has revenue at least $\texttt{Rev}(M_2)$.*

*Proof.* We only prove the first statement because the second one is analogous. We show that for each possible valuation $v \in V^{mn}$ (with $\|v_i\| \leq 1$ for all $i$) of the buyers, the total payment paid to the auctioneer for menu $M$ is at least as high as in menu $M_1$. Suppose for valuation $v \in V^n$ that the auctioneer chooses the $k(v)$-th option in menu $M_1$ in maximizing the boosted welfare. Note that we may assume $k(v) \in \mathcal{K}'_1$ due to 0-reducibility of $M_1$. By construction of $\varphi$, it follows that $\varphi_1(k(v)) = k(v)$. Therefore, the $k(v)$-th option in $M$ exactly equals the $k(v)$-th option in $M_1$. Because ties are broken in favor of larger total payments, it suffices to show that the $k(v)$-th option is the one with the highest boosted welfare also in $M$. [7]

Let $k' \in \mathcal{K}$ be an arbitrary index. The boosted welfare of option $k'$ in $M$ is

$$\sum_i v_i^\top (\lambda x_i^{(1,k')} + (1-\lambda)x_i^{(1,\varphi_1(k'))})) + (\beta^{(1,k')} + (1-\lambda)\beta^{(1,\varphi_1(k'))})$$

$$= \lambda(\sum_i v_i^\top x_i^{(1,k')} + \beta^{(1,k')}) + (1-\lambda)(\sum_i v_i^\top x_i^{(1,\varphi_1(k'))} + \beta^{(1,\varphi_1(k'))})$$

$$\leq \lambda(\sum_i v_i^\top x_i^{(1,k(v))} + \beta^{(1,k(v))}) + (1-\lambda)(\sum_i v_i^\top x_i^{(1,k(v))} + \beta^{(1,k(v))})$$

$$= \sum_i v_i^\top x_i^{(1,k(v))} + \beta^{(1,k(v))},$$

where the inequality follows because the $k(v)$-th option is boosted welfare maximizing for menu $M_1$. This shows that $k(v)$ is also a boosted welfare maximizer for menu $M$, completing the proof. $\square$

**Proposition 22.** *Let $M = \lambda \widehat{M}_1 + (1-\lambda)\widehat{M}_2$ be a convex combination of the menus $\widehat{M}_1$ and $\widehat{M}_2$. Then $\texttt{Rev}(M) \geq \lambda \texttt{Rev}(\widehat{M}_1) + (1-\lambda)\texttt{Rev}(\widehat{M}_2)$.*

*Proof.* The claim is trivial for $\lambda = 0$ or $\lambda = 1$. Therefore, assume $0 < \lambda < 1$ for the remainder of the proof. For possible valuation $v \in V^n$ such that $\|v_i\| \leq 1$ for all $i$, let $k_1(v)$ and $k_2(v)$ be the boosted welfare maximizing options in $\widehat{M}_1$ and $\widehat{M}_2$, respectively. Note that by the construction of the menus $\widehat{M}_1$ and $\widehat{M}_2$, we may assume without loss of generality that $k_1(v) \in \mathcal{K}'_1$ and $k_2(v) \in \mathcal{K}'_2$. Let $k^*(v) := \varphi^{-1}(k_1(v), k_2(v))$.

---

[7] Recall that the auctioneer will choose the option $k$ maximize the $\beta^{(k)}$ among all boosted welfare maximizing options given the formula of the total payment (2).

We show that option $k^*(v)$ is boosted welfare maximizing in $M = \{(x^{(k)}, \beta^{(k)})\}_{k \in \mathcal{K}}$ with valuation $v$. To this end, we use the notation $\widehat{M}_1 = \{(\hat{x}^{(1,k)}, \hat{\beta}^{(1,k)})\}_{k \in \mathcal{K}}$ and $\widehat{M}_2 = \{(\hat{x}^{(2,k)}, \hat{\beta}^{(2,k)})\}_{k \in \mathcal{K}}$. Let $k' \in \mathcal{K}$ be an arbitrary index. Then, the boosted welfare of option $k'$ can be bounded as follows:

$$\sum_i v_i^\top x_i^{(k')} + \beta^{(k')} = \lambda(\sum_i v_i^\top \hat{x}_i^{(1,k')} + \hat{\beta}^{(1,k')}) + (1-\lambda)(\sum_i v_i^\top \hat{x}_i^{(2,k')} + \hat{\beta}^{(2,k')})$$

$$\leq \lambda(\sum_i v_i^\top \hat{x}_i^{(1,k_1(v))} + \hat{\beta}^{(1,k_1(v))}) + (1-\lambda)(\sum_{ij} v_i \hat{x}_{ij}^{(2,k_2(v))} + \hat{\beta}^{(2,k_2(v))})$$

$$= \lambda(\sum_i v_i^\top \hat{x}_i^{(1,k^*(v))} + \hat{\beta}^{(1,k^*(v))}) + (1-\lambda)(\sum_i v_i^\top \hat{x}_i^{(1,k^*(v))} + \hat{\beta}^{(1,k^*(v))})$$

$$= \sum_i v_i^\top x_i^{(k^*(v))} + \beta^{(k^*(v))},$$

where the inequality in the second line follows because $k_1(v)$ and $k_2(v)$ are boosted welfare maximizers for $\widehat{M}_1$ and $\widehat{M}_2$, respectively. The equality in the third line follows because, by construction, in menu $\widehat{M}_1$ option $k^*(v)$ is equivalent to option $k_1(v) = \varphi_1(k^*(v)) \in \mathcal{K}_1'$. Similarly, in menu $\widehat{M}_2$ option $k^*(v)$ is equivalent to option $k_2(v) = \varphi_2(k^*(v)) \in \mathcal{K}_2'$. This concludes the proof that $k^*(v)$ is a boosted welfare maximizer in $M$.

Note that the total payment of $M = \{(x^{(k)}, \beta^{(k)})\}_{k \in \mathcal{K}}$ can be written in the following form:

$$\sum_i p_i(v) = \sum_i \left( \sum_{l \neq i} v_l^\top x_l^{(k(v_{-i}))} + \beta^{(k(v_{-i}))} \right) - \sum_i \left( \sum_{l \neq i} v_l^\top x_l^{(k(v))} + \beta^{(k(v))} \right).$$

where $k(\cdot)$ is the boosted welfare maximizer used in $M$. As ties are broken in favor of larger total payments, this value decreases by replacing $k(\cdot)$ by $k^*(\cdot)$ [8]:

$$\sum_i \left( \sum_{l \neq i} v_l^\top x_l^{(k(v_{-i}))} + \beta^{(k(v_{-i}))} \right) - \sum_i \left( \sum_{l \neq i} v_l^\top x_l^{(k(v))} + \beta^{(k(v))} \right)$$

$$\geq \sum_i \left( \sum_{l \neq i} v_l^\top x_l^{(k^*(v_{-i}))} + \beta^{(k^*(v_{-i}))} \right) - \sum_i \left( \sum_{l \neq i} v_l^\top x_l^{(k^*(v))} + \beta^{(k^*(v))} \right).$$

Since $k^*(v)$ is fixed for different $\lambda$ and, by linear combination, it holds that $\hat{x}^{(k^*(\cdot))} = \lambda \hat{x}^{(1,k^*(\cdot))} + (1-\lambda)\hat{x}^{(2,k^*(\cdot))}$ and $\hat{\beta}^{(k^*(\cdot))} = \lambda \hat{\beta}^{(1,k^*(\cdot))} + (1-\lambda)\hat{\beta}^{(2,k^*(\cdot))}$,

$$\sum_i \left( \sum_{l \neq i} v_l^\top x_l^{(k^*(v_{-i}))} + \beta^{(k^*(v_{-i}))} \right) - \sum_i \left( \sum_{l \neq i} v_l^\top x_l^{(k^*(v))} + \beta^{(k^*(v))} \right)$$

$$= \lambda \left[ \sum_i \left( \sum_{l \neq i} v_l^\top \hat{x}_l^{(1,k^*(v_{-i}))} + \hat{\beta}^{(1,k^*(v_{-i}))} \right) \right.$$

$$\left. - \sum_i \left( \sum_{l \neq i} v_l^\top \hat{x}_l^{(1,k^*(v))} + \hat{\beta}^{(1,k^*(v))} \right) \right]$$

$$+ (1-\lambda) \left[ \sum_i \left( \sum_{l \neq i} v_l^\top \hat{x}_l^{(2,k^*(v_{-i}))} + \hat{\beta}^{(2,k^*(v_{-i}))} \right) \right.$$

$$\left. - \sum_i \left( \sum_{l \neq i} v_l^\top \hat{x}_l^{(2,k^*(v))} + \hat{\beta}^{(2,k^*(v))} \right) \right]$$

$$= \lambda \texttt{Rev}(\widehat{M}_1) + (1-\lambda)\texttt{Rev}(\widehat{M}_2).$$

This completes the proof. $\qquad \square$

---

[8] Note that both $k(\cdot)$ and $k^*(\cdot)$ maximize the boosted welfare.

### D.2 Discretizing large menus

This subsection provides a detailed proof of Theorem 14. To do this, we will show that, for an AMA with a large number of options, one can discretize it such that, after discretization, the menu is 0-reducible. Additionally, during this discretization, the revenue loss will be up to $\varepsilon$.

**Lemma 23.** *Consider an AMA $M_1$ with at least $K + 1 = \lceil \frac{16m^3}{\epsilon^2} \rceil^{2nm}$ options. There exists an 0-reducible menu $\widetilde{M}_1$, such that, for any linear combination of $M_1$ and $\widetilde{M}_1$, $M = \lambda M_1 + (1-\lambda)\widetilde{M}_1$ for $\lambda \in [0,1]$, $\mathtt{Rev}(M) \geq \mathtt{Rev}(M_1) - \varepsilon$.*

Theorem 14 simply follows by combining Lemma 23 and Proposition 12.

Note that the payments and allocations only depend on those boosted welfare maximizing options. Therefore, to show that $\widetilde{M}_1$ is 0-reducible, it suffices to show $\widetilde{M}_1$ has at most $\sqrt{K+1}$ different allocations.

We now formally define $\widetilde{M}_1$. We introduce parameters $\tilde{\varepsilon}$ and $\delta$, which will be specified later.

**Construction of $\widetilde{M}_1$**    For $x^{(k)}$, we round it to $\tilde{x}^{(k)}$ in which $\tilde{x}_{ij}^{(k)} = \frac{\tilde{\varepsilon}}{m} \left\lfloor \frac{mx_{ij}^{(k)}}{\tilde{\varepsilon}} \right\rfloor$. With this rounding, for any $v \in [0,1]^{nm}$ such that $\|v_i\| \leq 1$ for $i$, [9]

$$\sum_i v_i^\top x_i^{(k)} \geq \sum_i v_i^\top \tilde{x}_i^{(k)} \geq \sum_i v_i^\top x_i^{(k)} - \tilde{\varepsilon}. \tag{6}$$

For $\beta^{(k)}$, we let $\tilde{\beta}^{(k)} = (1-\delta)\beta^{(k)}$.

**Lemma 24.** *For any given $0 < \varepsilon \leq \frac{1}{4}$, let $\delta = \frac{\sqrt{\tilde{\varepsilon}}}{m}$ and $\tilde{\varepsilon} = \frac{\varepsilon^2}{16m^2}$. Then,*

$$\mathtt{Rev}(\widetilde{M}_1) \geq \mathtt{Rev}(M_1) - \varepsilon.$$

*The number of different allocations in $\widetilde{M}_1$ is at most $\lceil \frac{16m^3}{\varepsilon^2} \rceil^{nm}$. Additionally, for any linear combination of $M_1$ and $\widetilde{M}_1$, $M = \lambda M_1 + (1-\lambda)\widetilde{M}_1$, $\mathtt{Rev}(M) \geq \mathtt{Rev}(M_1) - \varepsilon$.*

*Proof.* We first demonstrate $\mathtt{Rev}(\widetilde{M}_1) \geq \mathtt{Rev}(M_1) - m\tilde{\varepsilon} - \frac{m^2\delta}{1-\delta} - \frac{\tilde{\varepsilon}}{\delta}$. The result follows by picking $\delta = \frac{\sqrt{\tilde{\varepsilon}}}{m}$ and $\tilde{\varepsilon} = \frac{\varepsilon^2}{16m^2}$. The proof of the bound on the linear combination of $M_1$ and $\widetilde{M}_1$ is analogous. We use the notation $M_1 = \{x^{(k)}, \beta^{(k)}\}_{k \in \mathcal{K}}$ and $\widetilde{M}_1 = \{\tilde{x}^{(k)}, \tilde{\beta}^{(k)}\}_{k \in \mathcal{K}}$.

We fix the valuation $v$. Let $k(v) = \arg\max_k \sum_i v_i^\top x_i^{(k)} + \beta^{(k)}$ and satisfy the tie-breaking rule. The total payment using $M_1$ can be expressed as follows:

$$\sum_i \left( \sum_{l \neq i} v_l^\top x_l^{(k(v_{-i}))} + \beta^{(k(v_{-i}))} \right) - \sum_i \left( \sum_{l \neq i} v_l^\top x_l^{(k(v))} - \beta^{(k(v))} \right)$$

$$= \underbrace{\sum_i \left( \sum_{l \neq i} v_l^\top x_l^{(k(v_{-i}))} + \beta^{(k(v_{-i}))} \right)}_{A_1} - \underbrace{(m-1)\left( \sum_i v_i^\top x_i^{(k(v))} + \beta^{(k(v))} \right)}_{B_1} - \beta^{(k(v))}. \tag{7}$$

Similarly, let $\tilde{k}(v) = \arg\max_k \sum_i v_i^\top \tilde{x}_i^{(k)} + \tilde{\beta}^{(k)}$, then, the total payment with $\widetilde{M}_1$ is

$$\underbrace{\sum_i \left( \sum_{l \neq i} v_l^\top \tilde{x}_l^{(\tilde{k}(v_{-i}))} + \tilde{\beta}^{(\tilde{k}(v_{-i}))} \right)}_{A_2} - \underbrace{(m-1)\left( \sum_i v_i^\top \tilde{x}_i^{(\tilde{k}(v))} + \tilde{\beta}^{(\tilde{k}(v))} \right)}_{B_2} - \tilde{\beta}^{(\tilde{k}(v))}. \tag{8}$$

We bound the differences between $A_1$ and $A_2$, $B_1$ and $B_2$, and $\beta^{(k(v))}$ and $\tilde{\beta}^{(\tilde{k}(v))}$ separately.

---

[9]The second inequality holds as $v_i^\top(x_i^{(k)} - \tilde{x}_i^{(k)}) \leq \|v_i\|_1 \|x_i^{(k)} - \tilde{x}_i^{(k)}\|_\infty \leq \frac{\tilde{\varepsilon}}{m}$ for any $i$.

First, for the difference between $A_1$ and $A_2$, we can use the following inequalities: for any possible $v$ such that $\|v_i\| \le 1$ for all $i$,

$$\sum_i v_i^\top x_i^{(k(v))} + \beta^{(k(v))} \le \sum_i v_i^\top \tilde{x}_i^{(k(v))} + \tilde{\beta}^{(k(v))} + \tilde{\varepsilon} + \delta\beta^{(k(v))}$$

$$\le \sum_i v_i^\top \tilde{x}_i^{(\tilde{k}(v))} + \tilde{\beta}^{(\tilde{k}(v))} + \tilde{\varepsilon} + \delta\beta^{(k(v))}.$$

This implies

$$A_1 \le A_2 + m\tilde{\varepsilon} + \sum_i \delta\beta^{(k(v_{-i}))}. \tag{9}$$

Second, for the difference between $B_1$ and $B_2$, we can use the following inequalities: for any possible $v$ such that $\|v_i\| \le 1$ for all $i$,

$$\sum_i v_i^\top x_i^{(k(v))} + \beta^{(k(v))} \ge \sum_i v_i^\top x_i^{(\tilde{k}(v))} + \beta^{(\tilde{k}(v))}$$

$$\ge \sum_i v_i^\top \tilde{x}_i^{(\tilde{k}(v))} + \tilde{\beta}^{(\tilde{k}(v))} + \delta\beta^{(\tilde{k}(v))}.$$

This implies

$$B_1 \le B_2 - (m-1)\delta\beta^{(\tilde{k}(v))}. \tag{10}$$

Finally, we want to claim

$$\beta^{(\tilde{k}(v))} \le \beta^{(k(v))} + \frac{\tilde{\varepsilon}}{\delta}, \text{ which implies } \tilde{\beta}^{(\tilde{k}(v))} \le \beta^{(k(v))} + \frac{\tilde{\varepsilon}}{\delta} - \delta\beta^{(\tilde{k}(v))}; \tag{11}$$

as otherwise if $\beta^{(\tilde{k}(v))} > \beta^{(k(v))} + \frac{\tilde{\varepsilon}}{\delta}$, then this implies

$$\sum_i v_i^\top \tilde{x}_i^{(k(v))} + \tilde{\beta}^{(k(v))} \ge \sum_i v_i^\top x_i^{(k(v))} + \beta^{(k(v))} - \tilde{\varepsilon} - \delta\beta^{(k(v))}$$

$$\ge \sum_i v_i^\top x_i^{(\tilde{k}(v))} + \beta^{(\tilde{k}(v))} - \tilde{\varepsilon} - \delta\beta^{(k(v))}$$

$$\ge \sum_i v_i^\top \tilde{x}_i^{(\tilde{k}(v))} + \tilde{\beta}^{(\tilde{k}(v))} - \tilde{\varepsilon} + \delta(\beta^{(\tilde{k}(v))} - \beta^{(k(v))})$$

$$> \sum_i v_i^\top \tilde{x}_i^{(\tilde{k}(v))} + \tilde{\beta}^{(\tilde{k}(v))}, \tag{12}$$

which contradicts the fact that $\tilde{k}(v) = \arg\max_k \sum_i v_i^\top \tilde{x}_i^{(k)} + \tilde{\beta}^{(k)}$.

By combining the formula of total payment with $M_1$, (7), the formula of total payment with $\widetilde{M}_1$, (8), and inequalities (9), (10), (11); the loss on the total payment is at most $m\tilde{\varepsilon} + \sum_i \delta\beta^{(k(v_{-i}))} - m\delta\beta^{(\tilde{k}(v))} + \frac{\tilde{\varepsilon}}{\delta}$. Note that $\tilde{\beta}^{(k(v_{-i}))} \le \tilde{\beta}^{(\tilde{k}(v))} + m$[10]. Therefore, the total loss on the payment is at most $m\tilde{\varepsilon} + \frac{m^2\delta}{1-\delta} + \frac{\tilde{\varepsilon}}{\delta}$ which is $\text{Rev}(\widetilde{M}_1) \ge \text{Rev}(M_1) - m\tilde{\varepsilon} - \frac{m^2\delta}{1-\delta} - \frac{\tilde{\varepsilon}}{\delta}$.

Now, we prove a similar result for $M$, which is a linear combination of $M_1$ and $\widetilde{M}_1$: $M = \lambda M_1 + (1-\lambda)\widetilde{M}_1$. Let $M = (x'^{(k)}, \beta'^{(k)})_{k\in\mathcal{K}}$ and $k'(v) = \arg\max_k \sum_i v_i^\top x'^{(k)}_i + \beta'^{(k)}$. The total payment of $M$ is

$$\underbrace{\sum_i \left( \sum_{l\neq i} v_l^\top x'^{(k(v_{-i}))}_l + \beta'^{(k(v_{-i}))} \right)}_{A_3} \underbrace{- (m-1)\left( \sum_i v_i^\top x'^{(k(v))}_i + \beta'^{(k(v))} \right)}_{B_3} - \beta'^{(k(v))}. \tag{13}$$

---

[10]This is becasue $v^\top \tilde{x}^{(k(v_{-i}))} + \tilde{\beta}^{(k(v_{-i}))} \le v^\top \tilde{x}^{(\tilde{k}(v))} + \tilde{\beta}^{(\tilde{k}(v))}$ and $v^\top \tilde{x}^{(k(v))} \le m$.

Note that, by linear combination, for any $k$, $\sum_i v_i^\top x_i^{(k)} \geq \sum_i v_i^\top x_i'^{(k)} \geq \sum_i v_i^\top x_i^{(k)} - \tilde\varepsilon$ and $\beta'^{(k)} = (1 - \delta')\beta^{(k)}$ such that $\delta' = (1 - \lambda)\delta$. With similar proofs as (9) and (10), the following two inequalities hold,

$$A_1 \leq A_3 + m\tilde\varepsilon + \sum_i \delta'\beta^{(k(v_{-i}))}; \tag{14}$$

$$B_1 \leq B_3 - (m - 1)\delta'\beta^{(k'(v))}. \tag{15}$$

And, similarly,

$$\beta^{(k'(v))} \leq \beta^{(k(v))} + \frac{\tilde\varepsilon}{\delta}, \tag{16}$$

as otherwise $\beta^{(k'(v))} > \beta^{(k(v))} + \frac{\tilde\varepsilon}{\delta}$ implies

$$\sum_i v_i^\top x_i'^{(k(v))} + \beta'^{(k(v))}$$

$$= \lambda\left(\sum_i v_i^\top x_i^{(k(v))} + \beta^{(k(v))}\right) + (1 - \lambda)\left(\sum_i v_i^\top \tilde x_i^{(k(v))} + \tilde\beta^{(k(v))}\right)$$

$$> \lambda\left(\sum_i v_i^\top x_i^{(\hat k(v))} + \beta^{(\hat k(v))}\right) + (1 - \lambda)\left(\sum_i v_i^\top \hat x_i^{(k(v))} + \hat\beta^{(k(v))}\right)$$

$$= \sum_i v_i^\top x_i'^{(k'(v))} + \beta'^{(k'(v))}.$$

The strict inequality follows from (12) and $\lambda < 1$.

Therefore, combining inequalities (14), (15), (16), the total payment with $M_1$, (7), and the total payment with $M$, (13); the total loss for $M$ is at most $m\tilde\varepsilon + \frac{m^2\delta'}{1-\delta'} + \frac{\tilde\varepsilon}{\delta} \leq m\tilde\varepsilon + \frac{m^2\delta}{1-\delta} + \frac{\tilde\varepsilon}{\delta}$.

The result follows by picking $\delta = \frac{\sqrt{\tilde\varepsilon}}{m}$ and $\tilde\varepsilon = \frac{\varepsilon^2}{16m^2}$. $\qquad\square$

### D.3 Difference between softmax and argmax in revenue

Similar to the case of the *RochetNet*, in the training process, softmax operation is used instead of argmax. Recall that the revenue of AMA is

$$\texttt{Rev} = \int_v \sum_i p_i(v)\mathrm{d}F(v),$$

where

$$p_i(v) = \left(\sum_{l \neq i} v_l^\top x_l^{(k(v_{-i}))} + \beta^{(k(v_{-i}))}\right) - \left(\sum_{l \neq i} v_l^\top x_l^{(k(v))} + \beta^{(k(v))}\right),$$

and

$$k(v) = \arg\max_k \sum_i v_i^\top x_i^{(k)} + \beta^{(k)}.$$

For the softmax version, instead of using $k(v)$ which exactly maximizes the boosted social welfare, now $k^{\texttt{softmax}}(v)$ is a random variable:

$$k^{\texttt{softmax}}(v) = k \text{ with probability } \frac{e^{Y(v_i^\top x_i^{(k)} + \beta^{(k)})}}{\sum_{k'} e^{Y(v_i^\top x_i^{(k')} + \beta^{(k')})}};$$

and the price is the expectation on $k(v)$

$$p_i^{\texttt{softmax}}(v) = \mathbb{E}\left[\left(\sum_{l \neq i} v_l^\top x_l^{(k^{\texttt{softmax}}(v_{-i}))} + \beta^{(k^{\texttt{softmax}}(v_{-i}))}\right) \right.$$

$$\left. - \left(\sum_{l \neq i} v_l^\top x_l^{(k^{\texttt{softmax}}(v))} + \beta^{(k^{\texttt{softmax}}(v))}\right)\right];$$

and the revenue is

$$\text{Rev}^{\texttt{softmax}}(M) = \int_v \sum_i p_i^{\texttt{softmax}}(v) dF(v).$$

We show the following result. Note that we also assume the maximal density of a valuation type is $\mathcal{X}$.

**Theorem 25.**

$$|\textit{Rev}^{\textit{softmax}}(M) - \textit{Rev}(M)| \leq \frac{m(K+1)}{eY} + \frac{nm\mathcal{X}(K+1)}{Y}\left(1 + \log\frac{mY}{m\mathcal{X}}\right).$$

To prove this theorem, we need the following lemma which provides one of the basic properties of the softmax.

**Lemma 26.** *Given $L$ values, $a_1 \geq a_2 \geq a_3 \geq \cdots \geq a_L$, then $0 \leq a_1 - \sum_k a_k \frac{e^{Ya_k}}{\sum_{k'} e^{Ya_{k'}}} \leq \frac{L}{eY}$.*

*Proof.* It's clear that $0 \leq a_1 - \sum_k a_k \frac{e^{Ya_k}}{\sum_{k'} e^{Ya_{k'}}}$. On the other direction,

$$\begin{aligned}
a_1 - \sum_k a_k \frac{e^{Ya_k}}{\sum_{k'} e^{Ya_{k'}}} &\leq \sum_k (a_1 - a_k)\frac{e^{Y(a_k - a_1)}}{\sum_{k'} e^{Y(a_{k'} - a_1)}} \\
&\leq \frac{1}{Y}\sum_k Y(a_1 - a_k)\frac{e^{Y(a_k - a_1)}}{\sum_{k'} e^{Y(a_{k'} - a_1)}} \\
&\leq \frac{1}{Y}\sum_k e^{Y(a_1 - a_k) - 1}\frac{e^{Y(a_k - a_1)}}{\sum_{k'} e^{Y(a_{k'} - a_1)}} \\
&\leq \frac{L}{eY}\frac{1}{\sum_{k'} e^{Y(a_{k'} - a_1)}} \\
&\leq \frac{L}{eY}.
\end{aligned}$$ $\square$

Now, we can prove Theorem 25.

*Proof of Theorem 25.* We first give the upper bound on $\text{Rev}(M) - \text{Rev}^{\texttt{softmax}}(M)$.

Let $k(v) = \arg\max_k \sum_i v_i^\top x_i^{(k)} + \beta^{(k)}$ be the rule used in $\text{Rev}(M)$. Recall that

$$\sum_i p_i(v) = \sum_i \left(\sum_{l \neq i} v_l^\top x_l^{(k(v_{-i}))} + \beta^{(k(v_{-i}))}\right) - \sum_i \left(\sum_{l \neq i} v_l^\top x_l^{(k(v))} + \beta^{(k(v))}\right)$$

and

$$\begin{aligned}
\sum_i p_i^{\texttt{softmax}}(v) = \mathbb{E}\Bigg[ &\sum_i \left(\sum_{l \neq i} v_l^\top x_l^{(k^{\texttt{softmax}}(v_{-i}))} + \beta^{(k^{\texttt{softmax}}(v_{-i}))}\right) \\
&- \sum_i \left(\sum_{l \neq i} v_l^\top x_l^{(k^{\texttt{softmax}}(v))} + \beta^{(k^{\texttt{softmax}}(v))}\right)\Bigg].
\end{aligned}$$

Then,

$$\text{Rev}(M) - \text{Rev}^{\texttt{softmax}}(M)$$

$$= \mathbb{E}_v \left[ \sum_i \left( \sum_{l \neq i} v_l^\top x_l^{(k(v_{-i}))} + \beta^{(k(v_{-i}))} \right) \right.$$

$$- \mathbb{E}\left[ \sum_i \left( \sum_{l \neq i} v_l^\top x_l^{(k^{\texttt{softmax}}(v_{-i}))} + \beta^{(k^{\texttt{softmax}}(v_{-i}))} \right) \right]$$

$$- m \left( \sum_i v_i^\top x_i^{(k(v))} + \beta^{(k(v))} \right)$$

$$+ \mathbb{E}\left[ m \left( \sum_i v_i^\top x_i^{(k^{\texttt{softmax}}(v))} + \beta^{(k^{\texttt{softmax}}(v))} \right) \right]$$

$$\left. + \sum_i v_i^\top x_i^{(k(v))} - \mathbb{E}\left[ \sum_i v_i^\top x_i^{(k^{\texttt{softmax}}(v))} \right] \right]$$

$$\leq \frac{m(K+1)}{eY} + \mathbb{E}\left[ \sum_i v_i^\top x_i^{(k(v))} \right] - \mathbb{E}\left[ \sum_i v_i^\top x_i^{(k^{\texttt{softmax}}(v))} \right].$$

The inequality follows by Lemma 26. Then, we will bound the difference between $\mathbb{E}\left[ \sum_i v_i^\top x_i^{(k(v))} \right]$ and $\mathbb{E}\left[ \sum_i v_i^\top x_i^{(k^{\texttt{softmax}}(v))} \right]$. More specifically,

$$\mathbb{E}[v_{ij} x_{ij}^{(k(v))}] - \mathbb{E}[v_{ij} x_{ij}^{(k^{\texttt{softmax}}(v))}]$$

$$= \int_v v_{ij} x_{ij}^{(k(v))} - \sum_k v_{ij} x_{ij}^{(k)} \frac{e^{Y(\sum_{i'} v_{i'}^\top x_{i'}^{(k)} + \beta^{(k)})}}{\sum_{k'} e^{Y(\sum_{i'} v_{i'}^\top x_{i'}^{(k')} + \beta^{(k')})}} \mathrm{d}F(v)$$

$$\leq \int_v \sum_k (x_{ij}^{(k)} - x_{ij}^{(k(v))})^+ \frac{e^{Y(\sum_{i'} v_{i'}^\top x_{i'}^{(k)} + \beta^{(k)})}}{\sum_{k'} e^{Y(\sum_{i'} v_{i'}^\top x_{i'}^{(k')} + \beta^{(k')})}} \mathrm{d}F(v)$$

$$\leq \int_v \sum_k (x_{ij}^{(k)} - x_{ij}^{(k(v))})^+ e^{Y(\sum_{i'} v_{i'}^\top x_{i'}^{(k)} + \beta^{(k)} - \sum_{i'} v_{i'}^\top x_{i'}^{(k(v))} - \beta^{(k(v))})} \mathrm{d}F(v).$$

Recall that $(\cdot)^+ \triangleq \max\{\cdot, 0\}$. Let's define $\text{BW}(k) = \sum_{i'} v_{i'}^\top x_{i'}^{(k)} + \beta^{(k)}$ to be the boosted welfare of option $k$ for simplicity. Now, we focus on one option $k$, and we will give an upper bound on

$$\int (x_{ij}^{(k)} - x_{ij}^{(k(v))})^+ \mathbb{1}_{\text{BW}(k) + \sigma \geq \text{BW}(k(v)) \geq \text{BW}(k)} \mathrm{d}F(v) \qquad (17)$$

for the non-negative $\sigma$. The value of $\sigma$ will be determined later. Note that it is always true that $\text{BW}(k(v)) \geq \text{BW}(k)$ by the definition of $k(v)$. Additionally, if $\text{BW}(k) + \sigma \geq \text{BW}(k(v))$ is not satisfied then $e^{Y(\sum_{i'} v_{i'}^\top x_{i'}^{(k)} + \beta^{(k)} - \sum_{i'} v_{i'}^\top x_{i'}^{(k(v))} - \beta^{(k(v))})} \leq e^{-Y\sigma}$. Therefore, if (17) is upper bounded by $\mathcal{C}_{ij}(\sigma)$, then $\text{Rev}^{\texttt{softmax}}(M) - \text{Rev}(M) \leq \frac{mK}{eY} + (K+1)\left( \sum_{ij} \mathcal{C}_{ij}(\sigma) + nme^{-Y\sigma} \right)$.

Note that

$$\int (x_{ij}^{(k)} - x_{ij}^{(k(v))})^+ \mathbb{1}_{\text{BW}(k) + \sigma \geq \text{BW}(k(v)) \geq \text{BW}(k)} \mathrm{d}F(v)$$

$$= \int (x_{ij}^{(k)} - x_{ij}^{(k(v))})^+ \mathbb{1}_{\text{BW}(k(v)) - \text{BW}(K) \in [0, \sigma]} \mathrm{d}F(v). \qquad (18)$$

Now we fix all coordinates of valuation $v$ other than coordinate $ij$. Note that, the function $\text{BW}(k(v)) - \text{BW}(k) = \sum_{i'} v_{i'}^\top x_{i'}^{(k(v))} + \beta^{(k(v))} - \sum_{i'} v_{i'}^\top x_{i'}^{(k)} - \beta^{(k)}$ is a convex function on $v_{ij}$ and $(x_{ij}^{(k(v))} - x_{ij}^{(k)})$

is the gradient. Since we are looking at the region such that the function $\mathrm{BW}(k(v)) - \mathrm{BW}(k)$ is bounded in $[0, \sigma]$, this direct implies

$$\int (x_{ij}^{(k)} - x_{ij}^{(k(v))})^+ \mathbf{1}_{\mathrm{BW}(k)+\sigma \geq \mathrm{BW}(k(v)) \geq \mathrm{BW}(k)} \mathrm{d}F(v) \leq \mathcal{X}\sigma.$$

as the maximal density is at most $\mathcal{X}$. This implies $\mathrm{Rev}(M) - \mathrm{Rev}^{\texttt{softmax}}(M) \leq \frac{m(K+1)}{eY} + (K+1)nm\mathcal{X}\sigma + (K+1)nme^{-Y\sigma}$ which is upper bounded by $\frac{m(K+1)}{eY} + \frac{nm\mathcal{X}(K+1)}{Y}\left(1 + \log \frac{Y}{\mathcal{X}}\right)$ by setting $\sigma = \frac{1}{Y}\log \frac{Y}{\mathcal{X}}$.

The upper bound on $\mathrm{Rev}^{\texttt{softmax}}(M) - \mathrm{Rev}(M)$ follows by a similar argument. $\qquad \square$