# OpenReview forum: "Mode Connectivity in Auction Design"
_NeurIPS.cc/2023/Conference — NeurIPS 2023 poster_

### Official Review · Reviewer_xpkv · 2023-07-06

**Soundness:** 3 good
**Presentation:** 3 good
**Contribution:** 3 good
**Rating:** 7
**Confidence:** 2

**Summary:**

The paper seeks to provide a theoretical foundation for empirical results demonstrating that optimal auctions (both known and novel) can be discovered by differentiable auction theory.  It proves that two such auction formats satisfy a condition called ‘mode connectivity’: epsilon-mode connectivity is satisfied when two (local) solutions are connected by paths in parameter space.

**Strengths:**

**Originality**: to my knowledge, this work is novel.

**Quality**: this seems well executed.

**Clarity**: well-written.

**Significance**: the overall significance of the paper is hard to assess.  Insofar as it provides a theoretical basis for existing empirical results, I feel that the bar for significance is cleared.


**Weaknesses:**

My main concern about the paper is that it identifies a perplexing regularity - that both the RochetNet and AMA results require "five pieces" - but then does not seek to understand that.

Minor:
 - p.1: DSIC printed as DISC
 - l.79: “straight-jacket” or “strait-jacket”?
- l.135: “each in unit supply” clearer
 - l.166: whenever I see “it is … easy to see”, I want to see at least a hint of explanation: I’ve tried to cover over results that I suspected to be true (but later discovered weren’t) this way.


**Questions:**

 - Dominant strategy incentive compatibility is a strong concept.  I would appreciate a few sentences on what that buys them relative to e.g. Bayesian-Nash incentive compatibility.
 - l.66: re: the link between epsilon-dropout stability and epsilon-mode connectivity, is it useful to think in terms of a Lipschitz constant?
 - l.92 mentions an example with 10,000 neurons and 59 active options: can this be expressed in terms of K?
 - lines 126-127: why do the menu options define a “piecewise linear surface of utilities”?
 - My overall intuition from this is that large neural networks can overfit: a large number of neurons could, further, yield inactive options, which (as they are dominated?) can be thrown away without loss.  Is something like this correct?  If so, it would be worth emphasizing that.
 - My guess is that we need no more than one ‘contract’ per type of bidder: is this correct?  If so, it should be mentioned (if it hasn’t been).
 - To me, the most curious result is the “five pieces” element of some of the RochetNet and AMA theorems: what explains this?  How much more general is it?  (Thus, could the results of S3, S4 be presented as special cases?)  Can anything more specific be said about the pieces?


**Limitations:**

None.

---

> ### Author Rebuttal · Authors · 2023-08-09
>
> We thank the reviewer for carefully reading and assessing our paper and for the valuable feedback. We provide a detailed response to the raised issues / questions below:
>
> > p.1: DSIC printed as DISC
>
> A: Will be fixed, thanks!
>
> > l.79: “straight-jacket” or “strait-jacket”?
>
> A: This should be “straight-jacket”. We will add a reference for this.
>
> > l.135: “each in unit supply” clearer
>
> A: Will be added. Thanks!
>
> > l.166: whenever I see “it is … easy to see”, I want to see at least a hint of explanation: I’ve tried to cover over results that I suspected to be true (but later discovered weren’t) this way.
>
> A: The auction with a menu is DSIC, as the mechanism will provide the option with the buyer's maximal utility given the bid reported. Therefore, the buyer has no incentive to misreport.
>
> Questions:
>
> > Dominant strategy incentive compatibility is a strong concept. I would appreciate a few sentences on what that buys them relative to e.g. Bayesian-Nash incentive compatibility.
>
> A: For single buyer cases (i.e. RochetNet), Bayesian-Nash incentive compatibility is equivalent to dominant strategy incentive compatibility. They become different when there are multiple buyers (Affine Maximizer Auctions). For example, in the paper "Dominant-Strategy versus Bayesian Multi-item Auctions: Maximum Revenue Determination and Comparison" by Andrew Chi-Chih Yao, it is shown that these two concepts are different in the case when there are two buyers and two items. We want to emphasize that dominant strategy incentive compatibility is more robust than Bayesian one, as Bayesian-Nash incentive compatibility needs full knowledge of buyers' valuation distributions.
>
> > l.66: re: the link between epsilon-dropout stability and epsilon-mode connectivity, is it useful to think in terms of a Lipschitz constant?
>
> A: We are not exactly sure how to understand this question. Here is an attempt of an answer, please let us know if this is helpful: Lipschitz constant indicates that, for any point x, the function value does not change a lot within a neighborhood of x. However, when considering two distant points x and y which are far from each other, the Lipschitz constant does not provide a strong restriction on the function value along any path connecting x and y. It is possible that x and y are local minima of two valleys, and any path from x to y will encounter very high function values. In contrast, epsilon-mode connectiveity ensures this will not happen.
>
> > l.92 mentions an example with 10,000 neurons and 59 active options: can this be expressed in terms of K?
>
> A: Yes, K would be 10,000 in this example, and it seems that much fewer than all K menu options (namely only 59) are active.
>
> > lines 126-127: why do the menu options define a “piecewise linear surface of utilities”?
>
> A: We consider the relationship between valuation v and the utility u. According to the definition of RochetNet (see Section 2.1), $u = \max_k v^\top x^{(k)} - p^{(k)}$. This function u is a piecewise linear function in terms of v.
>
> > My overall intuition from this is that large neural networks can overfit: a large number of neurons could, further, yield inactive options, which (as they are dominated?) can be thrown away without loss. Is something like this correct? If so, it would be worth emphasizing that.
>
> A: From what we understand, your intuition is almost correct, and this is actually the main intuition behind Dughmi et al. [2014], which is the key tool in proving Theorem 9. However, one needs to be careful as it is possible that each neuron serves a very small region of valuation, and throwing these neurons may cause a large deviation of revenue. The way to deal with this is we also need to modify the remaining neurons after removing the others. Please let us know if this answers your questions.
>
> > My guess is that we need no more than one ‘contract’ per type of bidder: is this correct? If so, it should be mentioned (if it hasn’t been).
>
> A: Each buyer can only be assigned one option in the menu. However, even in the single-buyer case it might be crucial to offer the buyer many (for global optimality sometimes even infinitely many) options to choose from.
>
> > To me, the most curious result is the “five pieces” element of some of the RochetNet and AMA theorems: what explains this? How much more general is it? (Thus, could the results of S3, S4 be presented as special cases?) Can anything more specific be said about the pieces?
>
> A: The fact that our results work with five pieces stems from how our proofs work: we basically transform one menu into the other one in five linear steps. We do not know whether fewer than five pieces are possible, too. However, it also does not matter much: for us the most important message behind these five pieces is that it is a very simple structure: the path is not very complicated, it is just piecewise linear with very few pieces. It does not seem very relevant whether there are, say, 3, 5, or even 20 pieces.
>
> It is therefore not surprising, that both cases yield exactly 5 pieces: since the proof strategies are similar, the results are similar, too. It is possible that the results of S3 and S4 could both be derived as special cases of something more general, but we think that this more general framework, if it exists, would be very complicated and unintuitive. It is probably more valuable to have a clean, short, readible proof that only works for RochetNet, and a separate, more technical version for AMA, as we present it in this paper.

---

> > ### Comment · Reviewer_xpkv · 2023-08-17
> >
> > Thank you.  This reply maintains my conviction that this is publishable at NeurIPS.
> >
> > Very minor comments:
> > 1. I was being understated above: it really should be 'straitjacket'; q.v.  https://en.wikipedia.org/wiki/Straitjacket.  ('Strait' in this sense meaning 'narrow': e.g. the Straits of Gibraltar, Magellan, Malacca, Hormuz...)
> > 1. on line 92, with $k=10000$: I wanted to see the formula, in $k$, mapping to 59 active options.
> > 1. on offering infinitely many options, maybe I'm missing the obvious: from an economic theory point of view, we should only need to offer as many options as there are possible types.  Do we need to offer _more_ than that?  If so, this (I would guess) is related to the learning requirements of escaping from non-convexity?  If that's right, it could make sense to mention that the number of menu items partitions into requirements for efficiency/separating equilibria (e.g. one per type) and those for learning.
> > 1. on the five pieces present in each proof, I'm not convinced that this is just an artefact of your proof technique - although you may be right.  I'd love there to be a tighter reason, but accept that this may have to wait for subsequent papers.

---

> > > ### Author Response · Authors · 2023-08-18
> > >
> > > We thank the reviewer for the follow-up comments.
> > >
> > > 1. In fact, this concept is called straight-jacket auction and not straitjacket auction in previous auction design literature, see https://arxiv.org/abs/1404.2329, where this concept was introduced.
> > > 2. We understand the experiments by Dütting et al. (2019) more as qualitative justification for the assumptions of some of our theorems, and do not expect the number of active neurons to follow a precise formula in practice. Nevertheless, in order to be $\epsilon$-reducible with $K=10000$, the number of active neurons (with high probability) should be at most $\sqrt{K+1}=\sqrt{10001}$, which is approximately $100$. In the experiments by Dütting et al. only $59$ were active, which justifies that our assumption of $\epsilon$-reducibility is reasonable in practical cases.
> > > 3. In fact, even if our intuition might tell us otherwise, the optimal menu might require infinitely many options already in very simple settings with only a single bidder and only two different items (see Section 5.6 in Dütting et al. (2019)). This is a structural result and totally independent from any optimization algorithm or learning approach used.
> > > Nevertheless, your intuition about escaping from non-convexity by offering a large number of options is correct: this is exactly the idea behind our Theorems 9 and 13 and also seems to work in practice (compare once again the experiments by Dütting et al. (2019)).
> > > 4. We suppose this remains a mystery until someone solves it. ;-)

---

> > > > ### Comment · Reviewer_xpkv · 2023-08-21
> > > >
> > > > Thanks!  Your first point presents an interesting dilemma.
> > > >
> > > > p.14 of Giannakopoulos and Koutsoupias notes: "The SJA is the deterministic mechanism which satisfies the slice conditions for all dimensions as tightly as possible (hence its name)".
> > > >
> > > > Thus, they are describing a straitjacket: straight-jacket is an understandable mistake.
> > > >
> > > > The dilemma:
> > > > 1. do we continue to use Giannakopoulos and Koutsoupias' incorrect name?
> > > > 1. do we correct the SJA's inventors?

---

> > > > > ### Author Response · Authors · 2023-08-21
> > > > >
> > > > > Indeed an interesting dilemma. In fact, several follow-up works after Giannakopoulos and Koutsoupias use "straight-jacket", too. Thanks for bringing this to our attention and diving into it so carefully.

---

### Official Review · Reviewer_HRyy · 2023-07-07

**Soundness:** 3 good
**Presentation:** 1 poor
**Contribution:** 3 good
**Rating:** 5
**Confidence:** 5

**Summary:**

This paper studies the mode connectivity for specific neural network architecture, i.e., RochetNet and Affine Maximizer Auctions(AMA), where the local optimal solutions produce close local optimal solutions. To be specific,
- for linear utilities, $\epsilon$-reducible solutions implies $\epsilon$- mode connectivity.
- for $n$ items and linear utilities, for large menu options $K$, then $\epsilon$-mode connectivity holds between any solutions for any distribution.
- Similar results also holds for AMA.

**Strengths:**

- The problem studied is a heated topic, and is a really interesting interdisciplinary area between auction design and neural network design.
- The result is quite novel, and establish some difference against existing work on mode connectivity.

**Weaknesses:**

The network architecture difference of existing work and the neural network studied in this paper needs further clarification, and more emphasis.


The current presentation of this paper needs a major revision, especially on our contributions sections. To name a few:
- There are some concepts that refer to later section of the paper, and the reviewer strongly suggests moving these definitions (or create an informal version) to the introduction section for readability, e.g., $\epsilon$-reducible appeared for line 85, 89, 110, affine maximizer for line 103.
- line 73-83 doesn't include the results for this paper, but rather a brief intro of RochetNet.
- For reviewers unfamiliar with the word "options", it's really hard to understand the sentence in line 75. "single hidden layer where the neurons directly correspond to menu options".
- The reviewer strongly suggests to put figure 1 in the intro section, and adding more details to the figure would helps a lot of the understanding of the paper.
- In section 1.1, many results/theorems are mixed with existing results, which is quite confusing.
- The definition of mode connectivity doesn't appear until page 6 of the paper...

Some other minor comments:
- line 21, please replace "DISC" by "DSIC".
- line 79, "straight-jacket auction" should be supported by citations.


**Questions:**

- What's the definition of the word "option" studied in this paper?
- Does this $\epsilon$ scale with the upper bound of the distribution, of the distribution is not in $[0,1]$, but in $[0,H]?

**Limitations:**

This paper is theoretical, hence there are no experiment results.

---

> ### Author Rebuttal · Authors · 2023-08-09
>
> We thank the reviewer for carefully reading and assessing our paper and for the valuable feedback.
>
> We regret that the reviewer perceives the presentation of our paper as poor. We are thankful for the constructive feedback and will do our best to address the issues raised by the reviewer in the final version. We provide a detailed response below.
>
> > The network architecture difference of existing work and the neural network studied in this paper needs further clarification, and more emphasis.
>
> A: We explain these differences in lines 118 to 133 of the initial submission. We also provide formal definitions of the considered architectures in Sections 2.1 and 2.2. Nevertheless, we understand that we should emphasize these differences more (see also our response to reviewer aV1n), and will do so in the final version.
>
> > There are some concepts that refer to later section of the paper, and the reviewer strongly suggests moving these definitions (or create an informal version) to the introduction section for readability, e.g., $\varepsilon$-reducible appeared for line 85, 89, 110, affine maximizer for line 103.
>
> A: Thanks for the suggestions. We will move corresponding (informal) definitions to earlier points in the paper.
>
> > line 73-83 doesn't include the results for this paper, but rather a brief intro of RochetNet.
>
> A: We agree, but we feel that for a smooth reading flow, it is necessary to provide this brief intro here to understand our contribution. It is not long enough to be moved into a separate (sub)section. See also our explanation further down.
>
> > For reviewers unfamiliar with the word "options", it's really hard to understand the sentence in line 75. "single hidden layer where the neurons directly correspond to menu options".
>
> A: The word option (an allocation and a price offered to the buyer) is explained in the very same line after the colon. In the final version we will reformulate the sentence to make it more understandable. See also our response to the related question below.
>
> > The reviewer strongly suggests to put figure 1 in the intro section, and adding more details to the figure would helps a lot of the understanding of the paper.
>
> A: The purpose of Figure 1 is to illustrate the formal definition of the RochetNet. Going into such great detail in the intro already seems to be a bit over the top for us. If the reviewer insists, we can add another, simplified version of the figure to the intro in the final version.
>
> > In section 1.1, many results/theorems are mixed with existing results, which is quite confusing.
>
> A: Our impression is that providing the context (previous results) while presenting our own contributions is more important than a clear visual separation. We still carefully state which results are our own ones. If the reviewer insists, we can try to separate this more, but we fear that this makes it harder to understand the context.
>
> Our paper aims to provide a theoretical explanation for the existing empirical success of mechanism design. Therefore, to effectively convey our contribution, we believe it is necessary to first present the key details of the existing empirical results and introduce our theoretical findings subsequently.
>
> > The definition of mode connectivity doesn't appear until page 6 of the paper...
>
> A: It is defined informally in the mode connectivity paragraph. To make it visually more clear, we will convert it to a definition environment in the final version.
>
> Some other minor comments:
>
> > line 21, please replace "DISC" by "DSIC".
>
> A: Will be fixed. Thanks!
>
> > line 79, "straight-jacket auction" should be supported by citations.
>
> A: We will add a citation. Thanks!
>
> Questions:
>
> > What's the definition of the word "option" studied in this paper?
>
> A: An option is a pair consisting of an allocation and a price offered from the seller to the buyer. We define this in line 75 of the original submission. As stated above, we will reformulate this sentence in the final version to avoid confusion.
>
> A set of several options is called a menu. For example, in the two-items case, a menu with two (non-trivial) options could be as follows: the first option is to get the first item and pay 5 dollars. The second option is to get both items and pay 10 dollars. Furthermore, we always assume that the buyer also has the option to buy nothing and pay nothing. Out of all options, the buyer needs to choose a single one.
>
> > Does this scale with the upper bound of the distribution, of the distribution is not in $[0, 1]$, but in $[0,H]$?
>
> A: Yes, it does. Then, one needs to replace $\varepsilon$ with $H \varepsilon$.

---

> > ### Comment · Reviewer_HRyy · 2023-08-15
> > **Acknowledgement**
> >
> > I agree that the theoretical contribution of this paper is non-trivial, and this is not a special case of the existing mode connectivity literature. I updated my score accordingly, but I strongly suggest the author to pay detailed attention to the presentation.

---

### Official Review · Reviewer_hKdM · 2023-07-07

**Soundness:** 4 excellent
**Presentation:** 4 excellent
**Contribution:** 4 excellent
**Rating:** 8
**Confidence:** 4

**Summary:**

The starting point for the paper is recent work in the area of “differentiable economics”, in which high-revenue strategyproof auction mechanisms are found by optimizing parameters using machine-learning-inspired gradient descent techniques.

The authors consider the problem of selling multiple goods to a single buyer. In such setting, strategyproof mechanisms can always be identified with a “menu” of (allocation, price) pairs — the bidder chooses their best choice from the menu, and carefully optimizing the menu can increase revenue. The neural architecture that represents these menu items is called “RochetNet”. They also consider multi-buyer versions of this problem, in particular searching through the space of affine maximizer auctions (AMA), which are structurally similar (there is a “menu” of possible outcomes, and a “boost” which roughly plays the role of the price, although actual per-bidder payments are calculated according to a VCG-style rule).

In these classes of auctions, the performance goal (revenue) is a very non-convex function of the auction parameters. Nevertheless, first-order optimization seems to work well in finding good or even known-optimal mechanisms. Also, empirically, allowing the auctions to learn over thousands of menu items helped performance, even though at the end of training, only a handful of these menu items were actually used (and in some cases known-optimal mechanisms might only have 3-4 menu items). The authors of this paper aim to explain these phenomena.

They build on existing work in deep learning for more standard tasks, where things work similarly: even though loss landscapes are not convex, it is possible to optimize over them using first-order methods, and overparameterizing the neural networks seems to help with this. One recent direction of theoretical work aims to explain these phenomena using the concept of “mode connectivity” — there are various results showing that if it is possible to remove lots of neurons from a trained network and get almost the same performance, then any two such solutions must be connected by a continuous path where at any point along the path, the loss function is within some constant of the two solutions. Presumably, loss landscapes with such a property should be easier to optimize over.

The authors establish similar properties for the two types of auction architectures. In particular, between any two menus where there is a small subset of options that would be chosen by a bidder with probability 1-\epsilon, embedded in a much larger unused or redundant menu, they show that (epsilon) mode connectivity holds. Additionally, any pair of menus with sufficiently large (as a function of epsilon and the number of items) menus is also epsilon-mode connected. Analogous results hold for AMAs.This provides a nice theoretical explanation of why these auctions are unexpectedly easy to optimize successfully, and why using large menus may help optimization.

**Strengths:**

Two separate papers in this area observed an interesting and useful empirical phenomenon but gave no good explanation for it. This paper gives a very solid explanation, and is extremely interesting for that reason. It also will hopefully motivate better designs and learning techniques for strategyproof auction architectures.

**Weaknesses:**

Currently I don’t see any significant weaknesses.

**Questions:**

I recommend further discussion of Shen et al. “Automated mechanism design via neural networks”, which presents an architecture that for certain cases is equivalent to RochetNet. It’s cited at one point but the actual neural architecture merits discussion (it also encodes menu items).

Is there an explicit explanation of exactly why mode connectivity helps optimization? Maybe it’s out there in the literature already. If so I think it’s worth devoting a few sentences to this.

**Limitations:**

Limitations are adequately addressed — in particular, that these techniques probably won’t extend to even more complicated/flexible auction architectures — they rely on the relatively simple “menu based” or “list of allocations + boost” structure of the particular architectures in question.

---

> ### Author Rebuttal · Authors · 2023-08-09
>
> We thank the reviewer for carefully reading and assessing our paper and for the valuable feedback.
>
> > I recommend further discussion of Shen et al. “Automated mechanism design via neural networks”, which presents an architecture that for certain cases is equivalent to RochetNet. It’s cited at one point but the actual neural architecture merits discussion (it also encodes menu items).
>
> A: Thanks for pointing out this. We will incorporate further discussion of Shen et al. into our final version.  MenuNet, developed by Shen et al. also encodes menu items and it generalizes RochetNet as it allows non-linear utility functions and can incorporate other networks trained from interaction data. One difference between MenuNet and RochetNet is their handling of valuation distributions. RochetNet repeatedly samples the valuations from the underlying distribution, whereas MenuNet discretizes the buyer's valuation space to discrete values and treats all possible discrete valuations as a collective input.
>
> Of course, in those cases where RochetNet and MenuNet are equivalent, our results directly apply to MenuNet, too. However, for general, non-linear utilities, we believe that it is interesting to obtain similar results, but this might be much more challenging.
>
> > Is there an explicit explanation of exactly why mode connectivity helps optimization? Maybe it’s out there in the literature already. If so I think it’s worth devoting a few sentences to this.
>
> A: We will add such a discussion to the final version. Basically, the intuition why mode connectivity is desirable is the same as for previous mode connectivity results. In the following we outline our perspective on this.
>
> Mode connectivity can help to explain the empirical performance of stochastic gradient descent (sgd) (or ascent, in case of revenue maximization). To some extent, mode connectivity prevents a poor local minimal valley region on the function value, from which the sgd method cannot escape easily. Suppose such a bad local minimum exists. Then mode connectivity implies that there exists a path from this bad local minimum to a global minimum on which the loss function does not significantly increase. Therefore, the intuition is that from every bad local minimum, a (stochastic) gradient method would be able to find a way to escape.
>
> In fact, such a very bad local minimal valley region does appear in the auction setting when the menu size is 1 (see Appendix C), which seems to counter the empirical performance of achieving the global optimum eventually. This paper proves that this will not appear when the menu size is big.
>
> However, note that the mode connectivity cannot fully justify the success of gradient descent, and there is a gap here, which also exists in the previous work on mode connectivity for neural networks. Mode connectivity only suggests that the local search algorithms are unlikely to get completely trapped.

---

> > ### Comment · Reviewer_hKdM · 2023-08-21
> > **thanks**
> >
> > Thanks for responding to each of these questions.

---

### Official Review · Reviewer_aV1n · 2023-07-08

**Soundness:** 4 excellent
**Presentation:** 4 excellent
**Contribution:** 2 fair
**Rating:** 5
**Confidence:** 3

**Summary:**

This paper studies the mode connectivity in neural networks for design auction mechanisms. Specifically, the authors prove that locally optimal solutions are connected by a simple, piecewise linear path such that every solution on the path is almost as good as one of the two local optima.

**Strengths:**

The paper proposes to address a significant problem: how to design an auction mechanism via machine learning.

The proof is given in detail and looks all correct.

The presentation is clear and well-structured.

**Weaknesses:**

The novelty and significance are not clear. Is the mode connectivity in auction mechanism just a special case of the mode connectivity of neural networks in general? Following this intuition, the proof seems straightforward:

The functional from a neural network to its corresponding revenue is a bounded (continuous) functional; the functional from this neural network to its training loss is also a bounded functional. Thus, it is a continuous function from training loss to revenue. Therefore, the mode connectivity of neural networks in term of the training loss implies the mode connectivity of auction mechanism in term of revenue. Some may argue that these bounded/continuous properties may not always hold, but given the neural networks are usually trained by SGD or its variants, these properties should stand.

------

POST-REBUTTAL: my concerns are addressed.

**Questions:**

Please address the weakness above. I would like to increase the score if it can be cleared.

**Limitations:**

Please see the weaknesses above.

---

> ### Author Rebuttal · Authors · 2023-08-09
>
> We thank the reviewer for carefully reading and assessing our paper and for the valuable feedback. As we state in the paper already, we do not believe that mode connectivity in auction settings follows as a special case from general mode connectivity results. We will make this clearer in the final version.
>
> In response to the question raised by the reviewer, in the following, we explain (i) why we think the argument suggested by the reviewer is not sufficient to deduce mode connectivity for auction design from previous mode connectivity results and (ii) why we do not think that a similar reasoning is possible at all.
>
> (i) We are not sure what exactly the reviewer means by "training loss" in the suggested argument, but we suppose it is some loss function on some training data in a setting for which mode connectivity is known. We would like to point out that this setting is structurally very different from our setting, in which training data does not really come as labeled pairs. Instead, in the auction setting, the expected revenue is directly maximized via stochastic gradient ascent by sampling from the valuation distribution.
>
> Moreover, even if one were able to define a meaningful notion of "training loss" as suggested by the reviewer, for which mode connectivity holds by previous results and which depends continuously on the network parameters, the reasoning suggested by the reviewer does not work at two different points: 1. the revenue is NOT a continuous function of the network parameters due to the involved, discontinuous argmax. 2. Even if it was, there cannot exist a well-defined mapping from "training loss" to revenue (or vice versa): different values of revenue would correspond to the same values of "training loss" and vice versa. Therefore it does not even make sense to speak about continuity of such a mapping.
>
> (ii) More generally, deducing mode connectivity for auction settings from previous mode connectivity results appears to be infeasible for the following reason. To our best knowledge, previous works on the mode connectivity of neural networks crucially rely on the properties that the networks minimize a convex loss function between the predicted and actual values, and require linear transformations in the final layer. However, these properties do not hold in RochetNet and AMA networks, where the structures are fundamentally different. For example, the loss function of RochetNet is not even a function of the network's output, but is defined by choosing the price of the argmax option, which is neither a linear transformation nor convex. We do not see how previous techniques can be applied here with a simple modification.
>
> Actually, other than mode connectivity, many properties or characterizations of neural networks cannot be applied to auction settings. For example, we know, for neural networks, the optimal training loss tends to 0 with sufficient over-parameterization. In contrast, the optimal revenue of auction mechanisms is extremely challenging: it remains largely unknown for most buyers' valuation distributions, even in the RochetNet case.

---

> > ### Comment · Reviewer_aV1n · 2023-08-21
> >
> > Thanks for your response. Most of my concerns are cleared.

---

### Official Review · Reviewer_Cnd8 · 2023-07-25

**Soundness:** 4 excellent
**Presentation:** 4 excellent
**Contribution:** 3 good
**Rating:** 6
**Confidence:** 4

**Summary:**

This paper focuses on justifying the empirical success of this differentiable economics, particularly in the context of menu-based methods like RochetNet and Differentiable AMA auctions.

The authors introduce the concept of the $\epsilon$-mode connectivity property, which establishes that two locally optimal menus are connected by a simple path where the revenue loss along the path is at most $\epsilon$. The paper demonstrates that this property holds true under two conditions:
- If the valuations are normalized, and the menus are $\epsilon$-reducible, meaning that there exists a small subset of menus that are active for a buyer with a probability of $1-\epsilon$.
- If the number of menu options is sufficiently large.


**Strengths:**

**Significance:** Although menu-based approaches for differentiable economics show promising empirical performance, there is a lack of understanding regarding their theoretical properties. This paper takes the first stride towards exploring these theoretical aspects.

**Originality:**  While mode-connectivity has primarily been studied in the context of prediction problems involving convex losses with linear transformations in the final layer, RochetNet and Differentiable AMA minimize negated revenue loss, which involves more complex calculations. This fundamental difference sets them apart in their analysis.

**Clarity and Quality:** The paper is well written and easy to follow.

**Weaknesses:**

 - This paper would benefit a lot from a discussion on how exactly mode connectivity can be used to justify empirical performance. While I agree that this is a cool theoretical property, it is very unclear to me how this exactly would help
- There is also no discussion/comments what practitioners should make of this property at all - (are there any insights on how to initialize these networks better to break permutation symmetry of menus etc or anything that would help improve training)
- The optimal allocations are not necessarily finite (this assumption isn't state clearly wherever used). See Example 3 in [*]

[*] Daskalakis, C., Deckelbaum, A., and Tzamos, C. (2017). Strong duality for a multiple-good monopolist. Econometrica, 85:735–767.

**Questions:**

Please see Weaknesses.

---

> ### Author Rebuttal · Authors · 2023-08-09
>
> We thank the reviewer for carefully reading and assessing our paper and for the valuable feedback. In the following we respond to the three questions/weaknesses raised by the reviewer.
>
> >This paper would benefit a lot from a discussion on how exactly mode connectivity can be used to justify empirical performance. While I agree that this is a cool theoretical property, it is very unclear to me how this exactly would help
>
> A: We will add such a discussion to the final version. Basically, the intuition why mode connectivity is desirable is the same as for previous mode connectivity results. In the following we outline our perspective on this.
>
> Mode connectivity can help to explain the empirical performance of stochastic gradient descent (sgd) (or ascent, in case of revenue maximization). To some extent, mode connectivity prevents a poor local minimal valley region on the function value, from which the sgd method cannot escape easily. Suppose such a bad local minimum exists. Then mode connectivity implies that there exists a path from this bad local minimum to a global minimum on which the loss function does not significantly increase. Therefore, the intuition is that from every bad local minimum, a (stochastic) gradient method would be able to find a way to escape.
>
> In fact, such a very bad local minimal valley region does appear in the auction setting when the menu size is 1 (see Appendix C), which seems to counter the empirical performance of achieving the global optimum eventually. This paper proves that this will not appear when the menu size is big.
>
> However, we agree that the mode connectivity cannot fully justify the success of gradient descent, and there is a gap here, which also exists in the previous work on mode connectivity for neural networks. Mode connectivity only suggests that the local search algorithms are unlikely to get completely trapped.
>
> > There is also no discussion/comments what practitioners should make of this property at all - (are there any insights on how to initialize these networks better to break permutation symmetry of menus etc or anything that would help improve training)
>
> A: We see the main contribution of our paper in explaining the empirical success and providing theoretical foundations for already existent practical methods, and not in inventing new methods. Nevertheless, two insights a practitioner could use are as follows: (i) It is worth understanding the structure of the auction in question. If one can, e.g., understand whether $\epsilon$-reducibility holds for a particular auction, this might indicate whether RochetNet or AMA are good methods to apply to this particular case.
> (ii) Size helps: If one encounters bad local optima, increasing the menu size and rerunning RochetNet/AMA might be a potential fix (and will eventually lead to a network satisfying mode connectivity).
>
> > The optimal allocations are not necessarily finite (this assumption isn't state clearly wherever used). See Example 3 in [...].
>
> A: We agree, and we do mention this in our paper, e.g. in the second paragraph of the introduction: "Already for two items and a single buyer, the description of the optimal mechanism may be uncountable [...]." However, our statements are not missing any assumptions as our proofs do not require the optimal allocation to be finite. Even if the optimal allocation is infinite, mode connectivity between any two considered menus (stemming from RochetNet or AMA) holds under the stated assumptions.

---

> > ### Comment · Reviewer_Cnd8 · 2023-08-13
> > **Updated my scores**
> >
> > Thanks for the clarification. I have updated my scores - I hope the authors will add the required discussion in their final version.

---

### Decision · Program_Chairs · 2023-09-21

**Decision:**

Accept (poster)

**Comment:**

Recent work has shown that neural networks can be applied to learning optimal auction mechanisms, but their success has been largely demonstrated empirically. This paper attempts to provide a theoretical justification for the use of neural networks for auction design. They focus on a special case of menu-based auctions, and show that the neural network architecture in this case (dubbed as RochetNet) satisfies a property called "mode connectivity". This property has been shown in prior work to aid in better optimization using SGD.

All reviewers agree that the paper makes a solid technical contribution. Multiple reviewers, however, were concerned about the presentation not being clear, especially to an audience unfamiliar with auction design.

We strongly urge the authors to make the following changes to the camera-ready version:
- Make the introduction less technical and build intuition for what follows in the paper (Reviewer HRyy)
- Provide background material for a reader not entirely familiar with auction design and RochetNet (Reviewer HRyy)
- Explain clearly why mode connectivity is a useful property when applying SGD (Reviewers Cnd8, hKdM)
- Add a discussion on how a practitioner may benefit from mode connectivity (Reviewer Cnd8)

Please also incorporate other specific feedback the reviewers have provided.